# Adult-born granule cells improve stimulus encoding and discrimination in the dentate gyrus

Diego M Arribas[1], Antonia Marin-Burgin[1]*, Luis G Morelli[1,2,3]*

[1]Instituto de Investigacion en Biomedicina de Buenos Aires (IBioBA) – CONICET/ Partner Institute of the Max Planck Society, Polo Cientifico Tecnologico, Buenos Aires, Argentina; [2]Departamento de Fisica, FCEyN UBA, Ciudad Universitaria, Buenos Aires, Argentina; [3]Max Planck Institute for Molecular Physiology, Department of Systemic Cell Biology, Dortmund, Germany

**\*For correspondence:**
aburgin@ibioba-mpsp-conicet.
gov.ar (AM-B);
lmorelli@ibioba-mpsp-conicet.
gov.ar (LGM)

**Competing interest:** The authors declare that no competing interests exist.

**Abstract** Heterogeneity plays an important role in diversifying neural responses to support brain function. Adult neurogenesis provides the dentate gyrus with a heterogeneous population of granule cells (GCs) that were born and developed their properties at different times. Immature GCs have distinct intrinsic and synaptic properties than mature GCs and are needed for correct encoding and discrimination in spatial tasks. How immature GCs enhance the encoding of information to support these functions is not well understood. Here, we record the responses to fluctuating current injections of GCs of different ages in mouse hippocampal slices to study how they encode stimuli. Immature GCs produce unreliable responses compared to mature GCs, exhibiting imprecise spike timings across repeated stimulation. We use a statistical model to describe the stimulus-response transformation performed by GCs of different ages. We fit this model to the data and obtain parameters that capture GCs' encoding properties. Parameter values from this fit reflect the maturational differences of the population and indicate that immature GCs perform a differential encoding of stimuli. To study how this age heterogeneity influences encoding by a population, we perform stimulus decoding using populations that contain GCs of different ages. We find that, despite their individual unreliability, immature GCs enhance the fidelity of the signal encoded by the population and improve the discrimination of similar time-dependent stimuli. Thus, the observed heterogeneity confers the population with enhanced encoding capabilities.

## Editor's evaluation

The paper by Arribas et al. examines the coding properties of adult-born granule cells in the hippocampus at both the single cell and network level. This paper is of interest to the hippocampal and computational neuroscience fields because it provides a framework for understanding how adult-born granule cells in the hippocampus contribute to network processing. The paper contains interesting ideas, such as the analysis of input-output transformation by spike response models and the establishment of "greedy networks", and the conclusions drawn are supported by the data.

## Introduction

Cell diversity is ubiquitous in the brain and many studies have highlighted its importance for the encoding of stimuli by a population of neurons (*Shamir and Sompolinsky, 2006*; *Berry, 2018*; *Chelaru and Dragoi, 2008*; *Holmstrom et al., 2010*; *Marsat and Maler, 2010*; *Padmanabhan and Urban, 2010*; *Tripathy et al., 2013*; *Zeldenrust et al., 2021*). In the dentate gyrus of the hippocampus,

cell diversity is enhanced and structured by adult neurogenesis, a mechanism by which new neurons termed granule cells (GCs) are born beyond development (*van Praag et al., 2002*). These adult-born GCs mature in a stereotyped way, making a distinctive contribution to hippocampal function during maturation (*Danielson et al., 2016*; *Clelland et al., 2009*; *Nakashiba et al., 2012*; *Sahay et al., 2011*). After about 8 weeks, they become electrophysiologically indistinguishable from other mature neurons (*Laplagne et al., 2006*; *Mongiat et al., 2009*).

About 4 weeks after cell birth, immature GCs establish functional synapses that can activate post-synaptic targets in CA3 (*Toni et al., 2008*; *Gu et al., 2012*; *Temprana et al., 2015*). At this age, they also receive presynaptic inputs from within the hippocampus (*Vivar et al., 2012*), the septum (*Vivar et al., 2012*; *Ogando et al., 2021*), and the entorhinal cortex (*van Praag et al., 2002*; *Mongiat et al., 2009*; *Toni et al., 2007*; *Vivar et al., 2012*; *Marín-Burgin et al., 2012a*). Concurrently, the electrophysiological properties of immature GCs evolve continuously. Experiments in hippocampal slices have shown that 4-week-old GCs (4wGCs) exhibit different electrophysiological properties from those of mature GCs (mGCs), such as an increased excitation/inhibition balance, a higher membrane excitability, a slower membrane time constant, and lower action potential threshold (*Mongiat et al., 2009*; *Marín-Burgin et al., 2012a*; *Temprana et al., 2015*; *Pardi et al., 2015*). Therefore, around 4 weeks after cell birth, immature GCs become integrated into the hippocampal circuitry and could play a distinctive role in the encoding and transmission of information.

In vivo, immature GCs exhibit distinctive patterns of activity: they fire at higher rates, they are less spatially tuned, and are preferentially activated during spatial memory tasks (*Danielson et al., 2016*; *Kee et al., 2007*). Immature GCs also promote pattern separation, a computation usually associated with the dentate gyrus which presumably involves augmenting the differences between similar incoming activity before relaying it (*Leutgeb et al., 2007*). Behavioral experiments have shown that ablating or inhibiting immature GCs impairs pattern separation (*Danielson et al., 2016*; *Clelland et al., 2009*; *Marín-Burgin and Schinder, 2012b*), while enhancing neurogenesis or inhibiting mGCs' outputs improves it (*Nakashiba et al., 2012*; *Sahay et al., 2011*). Still, the mechanisms by which immature GCs' distinct properties shape these functional roles remain largely unexplored.

Unlike random cell diversity, neurogenesis introduces a stereotyped form of diversity, with well-defined maturational stages, that could be leveraged by the dentate gyrus network. Given their distinct properties, immature GCs could perform a differential encoding of incoming activity, promoting pattern separation and other functions. However, experimental studies that address questions related to coding are scarce for immature GCs, partly because recording from immature GCs in vivo is challenging. Pattern separation involves encoding two overlapping stimuli into dissimilar representations. Do mature and immature GCs encode stimuli differently? Furthermore, how do GCs of different age encode a stimulus into a spiking response? The diversity contributed by immature GCs could improve these representations, facilitating their discrimination.

Here, we record the membrane potential of individual GCs while they process fluctuating input currents in hippocampal slices. We use transgenic mice to label immature GCs of different ages and distinguish them from the population. First, we generate a single current stimulus template using a stochastic process with short temporal correlations. Then, we inject the same stimulus template multiple times into each GC to study the structure of the responses across time and trials. To investigate the encoding properties of GCs, we characterize the functional relationship between stimulus and response fitting a statistical model for each recorded GC. Using this encoding model to decode experimental and simulated responses, we study how well these responses preserve stimuli information content. We build populations of neurons in silico, by selecting from a pool of mature and immature neurons the ones that maximize the fidelity of the reconstruction. We use this approach to study the impact of age diversity on stimulus reconstruction and encoding. Finally, we use these populations in a pattern separation task to probe the contribution of immature GCs to the discrimination of time-varying stimuli.

## Results

## Immature GCs' responses are less reliable and less aligned with stimulation

Electrophysiological properties of immature GCs and mGCs are known to differ (*Mongiat et al., 2009*; *Yang et al., 2015*; *Marín-Burgin and Schinder, 2012b*). Thus, we wondered whether neurons of different age would produce responses with a different temporal structure to the same stimulus. To investigate this, we performed ex vivo whole cell recordings in mGCs (n=21) and adult-born GCs that were 4 and 5 weeks of age (4wGCs, n=22 and 5wGCs, n=20) (*Figure 1A*). Adult-born GCs of different ages were labeled using a Ascl1-CreERT2-Tom mice line (*Figure 1B*, Methods) (*Yang et al., 2015*). Brain slices were prepared 4 or 5 weeks after tamoxifen injection to obtain tomato expressing GCs of these ages. Excitatory and inhibitory synaptic transmission were blocked pharmacologically with kynurenic acid and picrotoxin, so that differences in recorded activity were only due to differences in the cells intrinsic properties. We measured the passive input resistance and time constant of the GCs by injecting small hyperpolarizing current steps at –70 mV. Both the input resistance and the time constant measured in this way decreased with age (*Figure 1—figure supplement 1*), consistent with previous reports (*Mongiat et al., 2009*; *Yang et al., 2015*). Additionally, when injected with depolarizing current steps, immature GCs fired at higher frequencies than mature ones for the same step amplitude (*Figure 1—figure supplement 1*).

Next, we used fluctuating stimuli that we designed from in vivo recordings of the intact network (*Pernía-Andrade and Jonas, 2014*), which show that GCs receive a wide range of frequencies, with a power spectrum that exhibits a power law decay (Methods). These types of stimuli produced responses with a rich and reproducible temporal structure (*Mainen and Sejnowski, 1995*). We generated a single stimulus template from an Ornstein-Uhlenbeck process with a short correlation time constant (*Figure 1C*, Methods). mGCs have smaller input resistances and integrate larger excitatory and inhibitory currents than immature GCs (*Figure 1—figure supplement 1*; *Mongiat et al., 2009*; *Marín-Burgin et al., 2012a*). Therefore, to make the firing rate of the responses comparable across GCs of different ages, we used the same stimulus template for all GCs while adapting its baseline and amplitude (*Figure 1—figure supplement 2*). We injected nine trials of the template stimulus in each GC and recorded the resulting membrane potentials. From these, we extracted the spike timings that we used to study the responses.

We first studied the alignment between the neural responses and the injected stimulus. For each GC, we smoothed the spike trains with a rectangular sliding window and averaged over trials to obtain a peri-stimulus time histogram (PSTH) (*Figure 1D*). We then computed the Pearson's cross-correlation between the injected stimulus and the PSTHs for different time lags (*Figure 1E*). The lag of the cross-correlation peak reflects the time scale of stimulus integration, and the peak value the degree of alignment between stimulus and response. The lag and the value at the peak of the cross-correlation were negatively and positively correlated with age respectively (*Figure 1F*). These results indicate that GCs' responses become faster with maturation and exhibit stronger alignment with the stimulus.

While spike times were often preceded by stimulus upswings, the previous analysis doesn't explore the variability in the responses across trials and different GCs. To investigate this, we introduced a coincidence ratio between pairs of recordings, each recording consisting of all trials (*Figure 1G*, Methods; *Paiva et al., 2009*; *Naud et al., 2011*). We defined the coincidence ratio as the average proportion of spike coincidences between all possible pairs of trials. A coincidence ratio of 0 means no coincidences between the recordings and a coincidence ratio of 1 means all spikes match their timing. We quantified the reliability in the response of single GCs by computing coincidences between pairs of different trials of the same GC.

The coincidence ratio spanned a wide range of values (*Figure 1H*). For each GC, we computed its average coincidence ratio with all other GCs of the same age (*Figure 1I*). Older pairs of GCs exhibited larger coincidence ratios, indicating their responses were more similar to each other. In addition, mixed-age pairs that contain 4wGCs have lower coincidence ratios than pairs of mature neurons alone (*Figure 1—figure supplement 2*). This could be related to individual immature GCs producing less reproducible responses, which can be quantified by the reliability. GC reliability, determined by the diagonal of *Figure 1H*, took generally higher values than the coincidence ratio as trials from a single GC were usually more similar to each other than to trials from a different GC. Reliability increased with maturation, taking values $0.29 \pm 0.03$ (mean ± 1 s.e.m.) for 4wGCs, $0.38 \pm 0.04$ for 5wGCs, and

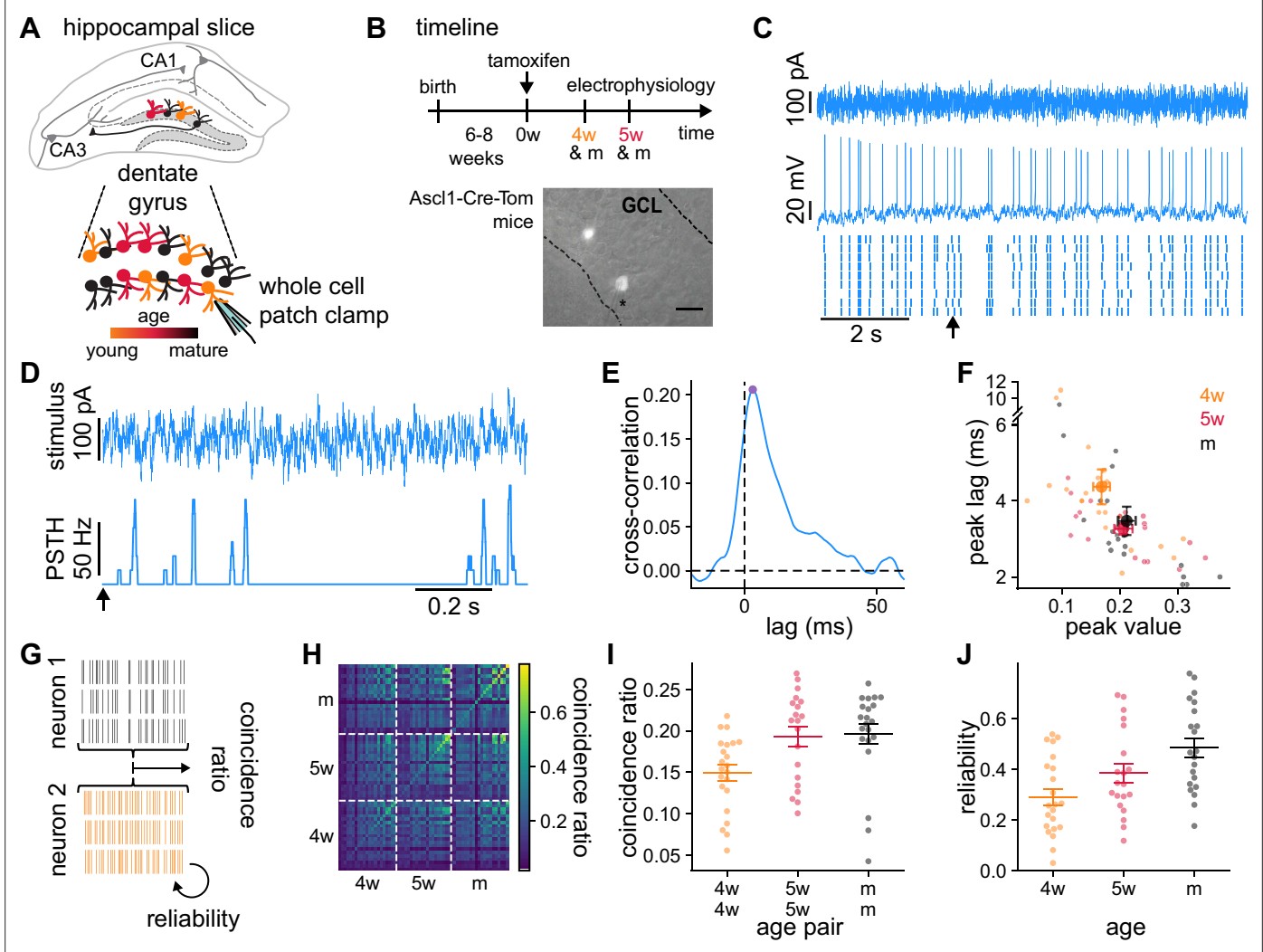

**Figure 1.** Granule cells (GCs) recordings and analysis of the temporal structure of the responses to the same stimulus. (**A**) Schematics of the experimental setup showing a hippocampal slice with the dentate gyrus highlighted in gray, and a blow-up of the dentate gyrus GCs. Colors indicate GCs' age: 4-week-old GCs (4wGCs) (orange), 5-week-old GCs (5wGCs) (red), and mature GCs (mGCs) (black). (**B**) Top: Experimental timeline. Tamoxifen is injected 6–8 weeks after mouse birth and slices are obtained 4 and 5 weeks after injection. Bottom: Hippocampal slice showing labeled immature GCs (bright spots) in the granule cell layer (GCL). Asterisk marks the electrode. Scale bar: 20 μm. (**C**) Recording of a mGC. Top: Fluctuating current stimulation. Middle: Recorded membrane potential from a single trial. Bottom: Spike raster plot showing nine trials obtained with the same stimulus. Arrow marks the starting time for panel (**D**). (**D**) Stimulus (top) and resulting peri-stimulus time histogram (PSTH) (bottom) from (**C**). (**E**) Cross-correlation between the stimulus and the PSTH of the mGC in (**C, D**). Cross-correlation peak (dot) is characterized by its peak value and lag. (**F**) Cross-correlation lag and peak value for all GCs recorded (small dots) and age group averages (large dots). Bars indicate mean ± 1 s.e.m. Spearman's correlation: $\rho = -0.30$, $p = 0.010$ between age and lag and $\rho = 0.25$, $p = 0.047$ between age and peak value. (**G**) The average fraction of coincident spikes defines (**i**) a reliability between different trials from a single GC and (**ii**) a coincidence ratio between all trials from different GCs. (**H**) Coincidence ratio matrix for all pairs of GCs. The diagonal is the reliability. (**I**) Average coincidence ratio between each GC and all other GCs of the same age (dots). Spearman's correlation with age of the pair: $\rho = 0.42$, $p = 7.1 \times 10^{-4}$. (**J**) Reliability of individual GCs (dots). Spearman's correlation with age: $\rho = 0.44$, $p = 2.7 \times 10^{-4}$. In (**I, J**) bars indicate mean ± 1 s.e.m. for each age group.

The online version of this article includes the following figure supplement(s) for figure 1:

**Figure supplement 1.** Intrinsic properties of granule cells (GCs) measured with current steps.

**Figure supplement 2.** Adjusting the baseline and amplitude of the stimulus to granule cells (GCs) of different ages while keeping the same time structure.

$0.49 \pm 0.04$ for mGCs (*Figure 1J*). Consistent with the cross-correlation analysis, this indicates that GCs are able to produce more robust responses with maturation. This effect could not be attributed to differences in firing rate, as shown by the reliabilities obtained after randomizing the timing of the spikes in every trial (*Figure 1—figure supplement 2*). Thus, our data indicates that immature GCs are less aligned with the stimulus and produce less reliable responses than mature ones.

## Immature GCs integrate over longer time scales and exhibit weaker refractory effects

The observed differences in the stimulus alignment and reliability of the responses suggest that GCs of different ages do not perform the same stimulus encoding. Thus, we sought to find a functional relationship between the stimulus and the GCs' responses to characterize their encoding properties, by fitting a spike response model (SRM) to our data (*Figure 2*; *Gerstner, 2014*). The SRM is a statistical model in the family of generalized linear models (GLMs) which allow for a direct fitting procedure and have been widely used to provide accurate statistical descriptions of spiking data (*Tripathy et al., 2013*; *Pillow et al., 2008*). A key feature of these models is that they provide a quantitative characterization of neural activity in terms of parameters that can be linked to biophysical properties.

The SRM describes both the subthreshold membrane potential and the spiking response to a current stimulation (*Figure 2A*, Methods). The stimulus first passes through a membrane filter $k(t)$ that defines how the input current is dynamically transduced in voltage variations. Next, the subthreshold membrane potential is generated by summing the filtered stimulus, a constant bias $v_b$, and the postspike voltage deflection $h_v(t)$. The bias is a baseline voltage and the postspike voltage deflection accounts for the effect of intrinsic currents occurring after a spike. A moving voltage threshold is then subtracted from the subthreshold membrane potential. The exponential of this difference scaled by a factor $\Delta v$ sets the time-dependent spiking rate. The moving threshold is the sum of a constant threshold $v_{th}$ and the postspike threshold deflection $h_{th}(t)$ that accounts for threshold refractory effects. Finally, spikes are randomly generated at every discrete time point from the time-dependent spiking rate.

We fitted the SRM to single trial 99 s recordings from individual 4wGCs (n=20), 5wGCs (n=17), and mGCs (n=18), using rapidly fluctuating stimuli generated as described in the previous section. We performed the SRM fitting in two steps: first we extracted the subthreshold parameters $v_b$, $k(t)$, and $h_v(t)$ using the subthreshold membrane potential (*Figure 2B–D*, Methods); then we extracted $v_{th}$, $h_{th}(t)$, and $\Delta v$ by maximizing the likelihood of the spike times while keeping the subthreshold parameters fixed (*Figure 2E–G*, Methods). The SRM yields predictions on both the subthreshold membrane potential and the spikes of the response. To validate the fits, we used the nine trials 10 s recordings described in the previous section. The SRM captured both the subthreshold membrane potential and spiking responses of the validation data (*Figure 2H*). We used the validation data to compute the root mean squared error (MSE) of the subthreshold membrane potential prediction and a normalized log-likelihood (*Figure 2—figure supplement 1*, Methods). To assess the quality of the spike trains generated by the SRM, we also computed the coincidence index $M_d$ between simulated and recorded spike trains (*Gerstner, 2014*; *Naud et al., 2011*). The reliability of the simulated spike trains was generally smaller than the reliability of the data but they were linearly correlated, indicating that the differences between GCs were mostly preserved (*Figure 2—figure supplement 1*).

The parameters obtained from the fit reflect differences in the way that GCs of different ages encode stimuli. While passive properties determine the membrane potential deflection for small current steps in the absence of spikes, the membrane filter $k(t)$ determines how the stimulation current is actively transduced into voltage during spiking activity. $k(t)$ is composed of a rapid decay attributed to the patch pipette (*Pozzorini et al., 2015*) and a tail that is well approximated by an exponential decay (*Figure 2C*). The time constant of the slower exponential decay measures the time scale of stimulus integration. The area under the exponential is an electrical resistance and determines the average membrane potential deflection in the absence of spikes. Both the population time constant and the resistance of the filter $k(t)$ decreased with age (*Figure 2I*, *Figure 2—figure supplement 1*), consistent with the passive properties' observations (*Figure 1—figure supplement 1*; *Mongiat et al., 2009*; *Yang et al., 2015*). These properties were correlated with their passive counterparts but were generally smaller (*Figure 2—figure supplement 1*). Slower membranes could partly explain the observed unreliability of immature GCs, as neurons with longer time constants tend to be less reliable

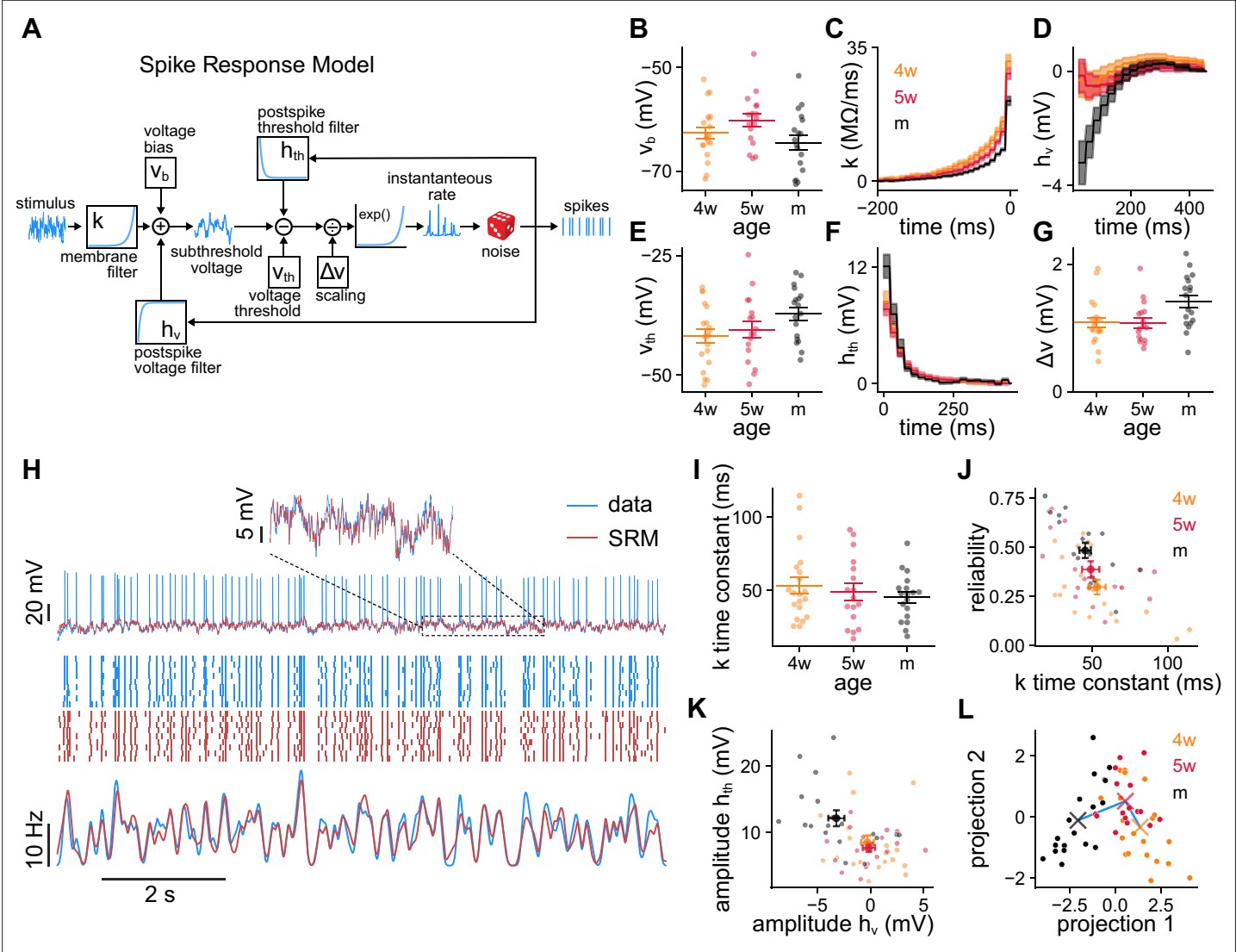

**Figure 2.** Spike response model (SRM) fits to recorded granule cells (GCs). (**A**) Schematics of the SRM. Free model parameters are highlighted within boxes. (**B–G**) SRM parameters obtained for all GCs of different ages: (**B**) voltage bias $v_b$, (**C**) membrane filter $k(t)$, (**D**) postspike membrane potential deflection $h_v(t)$, (**E**) static voltage threshold $v_{th}$, (**F**) postspike threshold deflection $h_{th}(t)$, and (**G**) voltage scaling factor $\Delta v$. (**H**) Validation data (blue) and SRM prediction (red). Top: Subthreshold membrane potential. Middle: Spike raster plots of the recorded responses and SRM simulations. Bottom: Peri-stimulus time histogram (PSTHs) of the spike trains. (**I**) Time constants extracted from the tail of filter $k(t)$. Spearman's correlation between age and population time constant (bootstrapped): $\rho = -0.567$, $p = 6.3 \times 10^{-2}$. (**J**) GC reliability vs. time constant of the filter $k(t)$. Spearman's correlation: $\rho = -0.61$, $p = 6.4 \times 10^{-7}$. (**K**) Amplitudes of the filters $h_v(t)$ and $h_{th}(t)$. Spearman's correlation: $\rho = -0.37$, $p = 6.0 \times 10^{-3}$ between age and $h_v^1$, and $\rho = 0.35$, $p = 8.1 \times 10^{-3}$ between age and $h_{th}^1$. (**L**) GCs' parameters projected on the linear discriminant analysis components subspace. The crosses represent the means of the age groups, connected by a blue line. In (**B, E, G, I–L**) small dots represent single neurons. In (**B, E, G, I**) long bars indicate means and short bars means ± 1 s.e.m. In (**J, K**) large dots and bars indicate means ± 1 s.e.m. In (**C, D, F**), lines with shaded areas indicate means ± 1 s.e.m.

The online version of this article includes the following figure supplement(s) for figure 2:

**Figure supplement 1.** SRM fitting validations.

**Figure supplement 2.** Linear discriminant analysis components as determined by the scalings of each parameter used.

---

(*Figure 2J*). The filters $h_v(t)$ and $h_{th}(t)$ quantify the postspike deflections in membrane potential and spiking threshold respectively. We quantified their amplitudes by using the first values of the filters (*Figure 2K*). While mGCs' membranes tended to hyperpolarize after a spike, this effect was generally smaller for immature GCs (*Figure 2D, K*). This hyperpolarization could be attributed to spike-triggered potassium currents, which have been reported to be smaller in immature GCs (*Mongiat*

*et al., 2009*; *Yang et al., 2015*). While the threshold after a spike increased for all GCs, the amplitude of the filter $h_{th}(t)$ was also positively correlated with age.

Next, we wanted to explore whether the obtained parameters could be used to discriminate GCs' ages. We used $v_b$, $v_{th}$, $\Delta v$, and the first coefficients of the filters $k(t)$, $h_v(t)$, and $h_{th}(t)$ to classify GCs' ages using a linear discriminant analysis (LDA) (*Figure 2L*; *Bishop, 2006*). SRM parameters could be used to classify mGCs and 4wGCs with cross-validated accuracies of 0.72 and 0.75, respectively. The other 0.28 and 0.25 fractions were classified as 5wGCs indicating that 4wGCs and mGCs were very well discriminated by the LDA. Being maturationally in between, the classification accuracy of 5wGCs was 0.47, with a fraction of 0.41 5wGCs being classified as 4wGCs. This suggests that 5wGCs were closer in parameter space to 4wGCs than to mGCs. The coefficients of the filter $k(t)$ play an important role in the classification as they have large weights in the LDA components (*Figure 2—figure supplement 2*). The LDA indicates that the SRM parameters reflect maturational differences and can be used to distinguish GCs' ages with higher than chance accuracy.

We observed that mGCs have a smaller input resistance, a faster membrane, and experience stronger postspike refractory effects. Our results suggest that GCs of different ages might transmit different properties of an afferent stimulus. Besides providing us with parameters that can be compared across ages, the SRM provides us with a statistical description of the GCs. This description can be used to generate spike trains and further explore the way in which GCs represent stimuli. Next, we use the SRM to generate simulated granule cells (SGCs) in a decoding framework to explore the fidelity of stimulus encoding and quantify the information content in the spike trains of the different age groups.

## Immature GCs transmit less information per spike and produce less precise reconstructions of stimuli

As spiking responses encode information about the stimulus, we wondered, what can we infer about the stimulus by observing the responses of GCs of different ages? To investigate this question, we used the SRM to perform Bayesian model-based decoding (*Tripathy et al., 2013*; *Pillow et al., 2008*; *Pillow et al., 2011*). Model-based decoding finds the most probable stimulus that produced one or more spike trains, given a prior distribution (*Figure 3A*, Methods). This decoded stimulus can then be compared with the original to evaluate the fidelity of the reconstruction. Furthermore, the procedure can be used to estimate the mutual information between stimulus and responses (*Pillow et al., 2011*). More broadly, we can use this framework to asses the fidelity of the code, that is to determine how well GCs of different ages encode information about the stimulus in the spike trains they produce.

We first performed the decoding using single experimentally recorded spike trains. We found the most probable stimulus that produced the spikes and the uncertainty about its value as a function of time (*Figure 3B*). During intervals in which a GC is silent, there is no information about the stimulus; hence the estimation takes baseline values and the uncertainty is highest. Just before an isolated spike, the stimulus is usually predicted to be above average and the uncertainty is reduced. Averaging over spikes and GCs, we found that the minimum uncertainty about the stimulus before a spike was reduced more in mGCs (*Figure 3C*).

While recorded spike trains allowed us to perform the decoding of the experimental stimulus, the SRM can be used to simulate responses to multiple different stimuli for decoding. Thus, we obtained SGCs by sampling different stimuli from the same stochastic process that was used experimentally, and repeated the decoding procedure using single spike trains generated with the SRM parameters of each GC. This strategy allows us to quantify decoding performance by estimating the mean coefficient of determination $r^2$ of the reconstruction, and the mutual information between the stimulus and the SGC spike trains (Methods). The coefficient $r^2$ involves computing the error between the reconstructed and the original stimuli to obtain the fraction of the variance explained by the decoded signal. In other words, this quantity captures the proportion of the variation in the stimulus that is predictable with the decoding. The mutual information measures the correspondence between stimulus and response, and quantifies the reduction in uncertainty about the stimulus after observing the given spike trains (*Schneidman et al., 2003*). Both the total $r^2$ and mutual information strongly correlated with SGC firing rate (*Figure 3D, E*). This is expected as we used only the spikes to decode the stimulus. Decoding performance using multiple stimuli and SGCs' spike trains was very similar to the one obtained by using the experimentally recorded stimuli and GCs' spike trains (*Figure 3—figure supplement 1*). To study the difference between GCs' ages, we removed the average contribution of

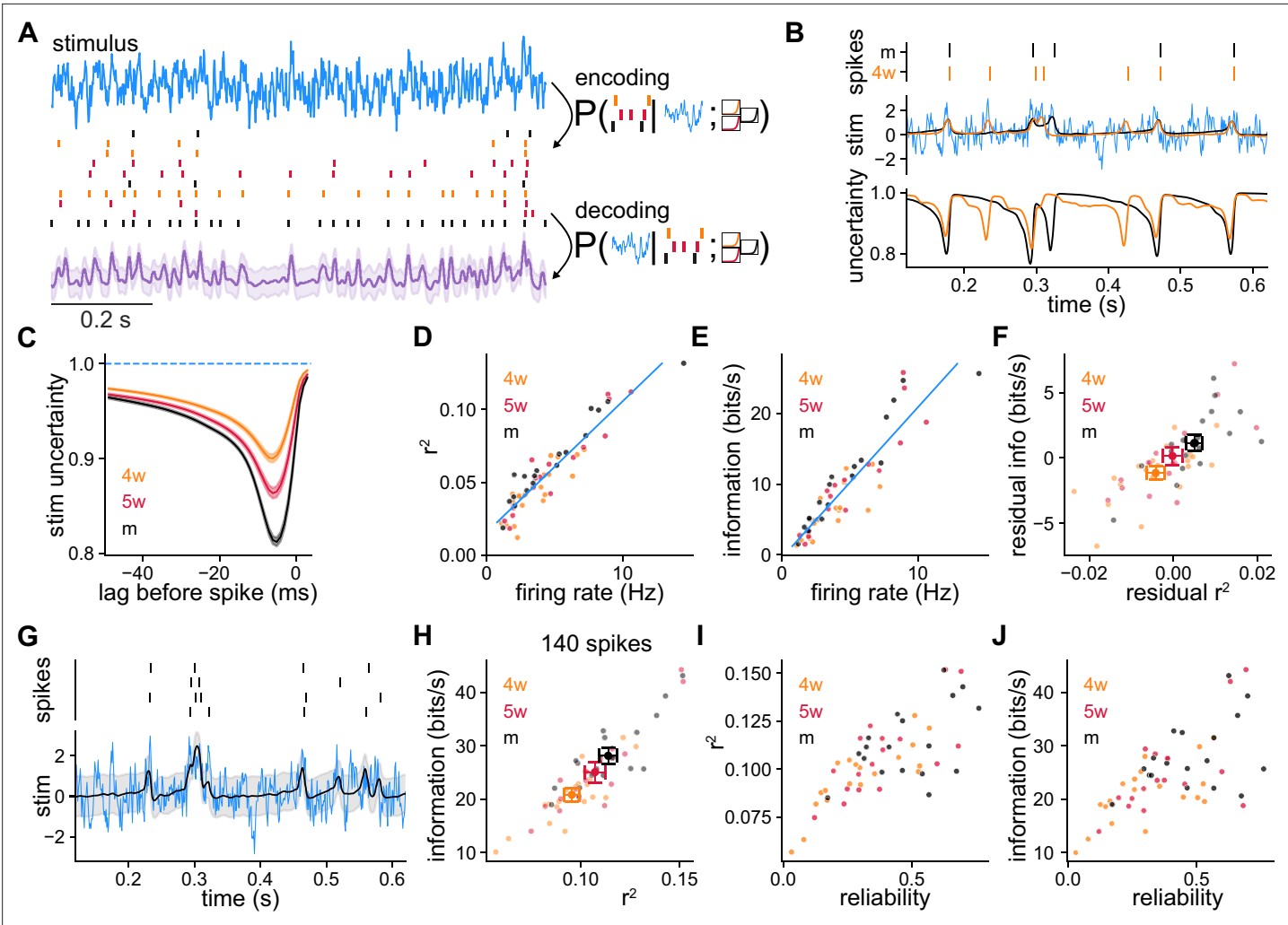

**Figure 3.** Model-based Bayesian decoding of the stimulus using recordings of single granule cells (GCs) and simulated GCs (SGCs). (**A**) Model-based Bayesian decoding scheme, illustrating how CGs or SGCs encode a stimulus in spike trains that can be used to estimate the stimulus that produced them. (**B**) Spike trains from recorded GCs (top) used separately to obtain two stimulus estimations (middle) and their respective uncertainties (bottom) about the stimulus. Mature GC (mGC) decoding in black ($r^2 = 0.058$), 4-week-old GC (4wGC) decoding in orange ($r^2 = 0.073$), true stimulus in blue. (**C**) Average uncertainty about the stimulus before a spike at lag 0, for recorded GCs. Spearman's correlation between minimum uncertainty and age: $\rho = -0.51$, $p = 6.0 \times 10^{-5}$. (**D**) Coefficient of determination $r^2$ obtained by decoding with individual spike trains vs. firing rate, from SGCs. Pearson's correlation: $\rho = 0.93$, $p = 3.6 \times 10^{-25}$. The blue line is the linear fit using all SGCs. (**E**) Estimated mutual information between the stimulus and SGCs' responses vs. firing rate. Pearson's correlation: $\rho = 0.91$, $p = 2.4 \times 10^{-22}$. The blue line is the linear fit using all SGCs. (**F**) Residual information vs. residual $r^2$ after subtracting the linear relationships of (**D, E**). Spearman's correlation between age and residual $r^2$: $\rho = 0.39$, $p = 2.9 \times 10^{-3}$; Spearman's correlation between age and residual information: $\rho = 0.38$, $p = 3.8 \times 10^{-3}$. (**G**) Decoding example using spike trains from a single SGC produced by multiple trials of the same stimulus ($r^2 = 0.11$). (**H**) Information vs. $r^2$ obtained by decoding with 140 spikes on average using a different number of trials for each SGC to compensate for firing rate differences. Spearman's correlation between age and $r^2$: $\rho = 0.37$, $p = 6.0 \times 10^{-3}$; Spearman's correlation between age and information: $\rho = 0.45$, $p = 6.1 \times 10^{-4}$. (**I**) $r^2$ from (**H**) vs. SGC reliability. Spearman's correlation: $\rho = 0.71$, $p = 1.2 \times 10^{-9}$. (**J**) Information from (**H**) vs. SGC reliability. Spearman's correlation: $\rho = 0.62$, $p = 5.1 \times 10^{-7}$.

The online version of this article includes the following figure supplement(s) for figure 3:

**Figure supplement 1.** Decoding the experimentally used stimulus from the recorded spike trains.

**Figure supplement 2.** Decoding performance with increasing number of trials.

**Figure supplement 3.** Decoding with pairs of SGCs of different age groups.

firing rate to decoding performance. For this, we subtracted the linear relationships of *Figure 3D, E* obtained with all SGCs from the $r^2$ and information of each SGC, obtaining the respective residuals (*Figure 3F*). Both the residual $r^2$ and information correlated with GC age indicating that for a given firing rate, responses from mGCs generally produced more precise reconstructions and were more informative about the stimulus.

We showed that GCs have different degrees of reliability and the same stimulus will elicit different individual responses from the same neuron (*Figure 1C*). Is it possible to extract more information from a single GC by observing multiple responses to a single stimulus? Does GC reliability influence this? Using responses to multiple trials of the same stimulus improved both the $r^2$ and the information of the decoding (*Figure 3G*). For small fold increments in the number of trials used for decoding, the information about the stimulus increased by almost the same fold (*Figure 3—figure supplement 2*). Immature SGCs benefit more from using multiple trials for decoding as a consequence of their noisier responses. To study age differences while equalizing differences in firing rate, we decoded stimuli using for each SGC a number of trials such that the total number of spikes was 140, on average (*Figure 3H*, Methods). Consistent with the results of *Figure 3F*, when equalizing the number of spikes for each SGC, decoding with immature SGCs resulted in lower $r^2$ and information values. When using larger numbers of spikes to equalize the firing rate differences, the correlations between age and decoding performance decreased (*Figure 3—figure supplement 2*). We also found that when equalizing the number of spikes used for decoding, the resulting $r^2$ and information values were larger in the SGCs that produced more reliable responses (*Figure 3I, J*).

We next studied how the different age groups complement each other by using pairs of different SGCs to decode. We used a single stimulus to generate 140 spikes on average from each SGC in the pair and performed the decoding (*Figure 3—figure supplement 3*). Using pairs of SGCs of the same age, we found that more mature pairs achieved higher values of $r^2$ and information. Pairing mature SGCs (mSGCs) with immature SGCs tended to degrade decoding performance resulting in smaller $r^2$ and information values. Coherent with the results from single SGCs, more immature pairs achieved overall worse decoding performance.

Our results indicate that immature GCs produce less precise reconstructions and convey less information about the stimulus, suggesting that they produce generally worse stimuli representations. Moreover, mGCs mostly increase decoding performance when paired with other mGCs rather than with immature ones. Thus, these results cast doubts about whether immature GCs can enhance coding in a population.

## Immature GCs improve stimulus reconstruction in a population

Our results so far suggest that mGCs, isolated or in pairs, are generally better than immature GCs for reconstructing stimuli. However, in a larger population there might be synergistic effects, and GCs that did not perform well by themselves could still play a significant role in enhancing population coding. Thus, we wondered whether immature GCs could improve decoding in a population consisting of a larger group of neurons. When building a population, even for a moderate number of neurons, trying all possible combinations between them results in prohibitive computational costs. Hence, we adopted a sequential greedy procedure to construct populations of SGCs that optimize stimulus reconstruction (*Tripathy et al., 2013*). To build a population, we started with a single mSGC and sequentially added individual neurons from the full pool of mature (n=18) and immature (5wSGCs n=17, 4wSGCs n=20) SGCs in the simulated experiment (*Figure 4A*, Methods). At each step, using the group built so far, we performed the decoding adding each possible SGC to the population, and kept the one that yielded the largest $r^2$ for the extended group. With this greedy procedure, we sought to evaluate whether immature neurons may contribute to improve the decoding, despite having lower $r^2$ and information values individually. Thus, we did not constrain the proportion of immature neurons in the population to match the in vivo estimation of 3% (*van Praag et al., 2002*). Rather, we always picked from these given pools the neuron that maximized the population $r^2$. As decoding performance depends on the number of spikes (*Figure 3D*), we used a different number of trials for each SGC so that the total number of spikes that each one contributed was 1200 on average (*Figure 4B*). In this way, we reduced the preference for trivially choosing SGCs with high firing rates.

We built mixed-age populations following the greedy procedure. In the first step, we started with the 18 mSGCs obtained with the SRM fit (*Figure 4C*). As expected from the results decoding with

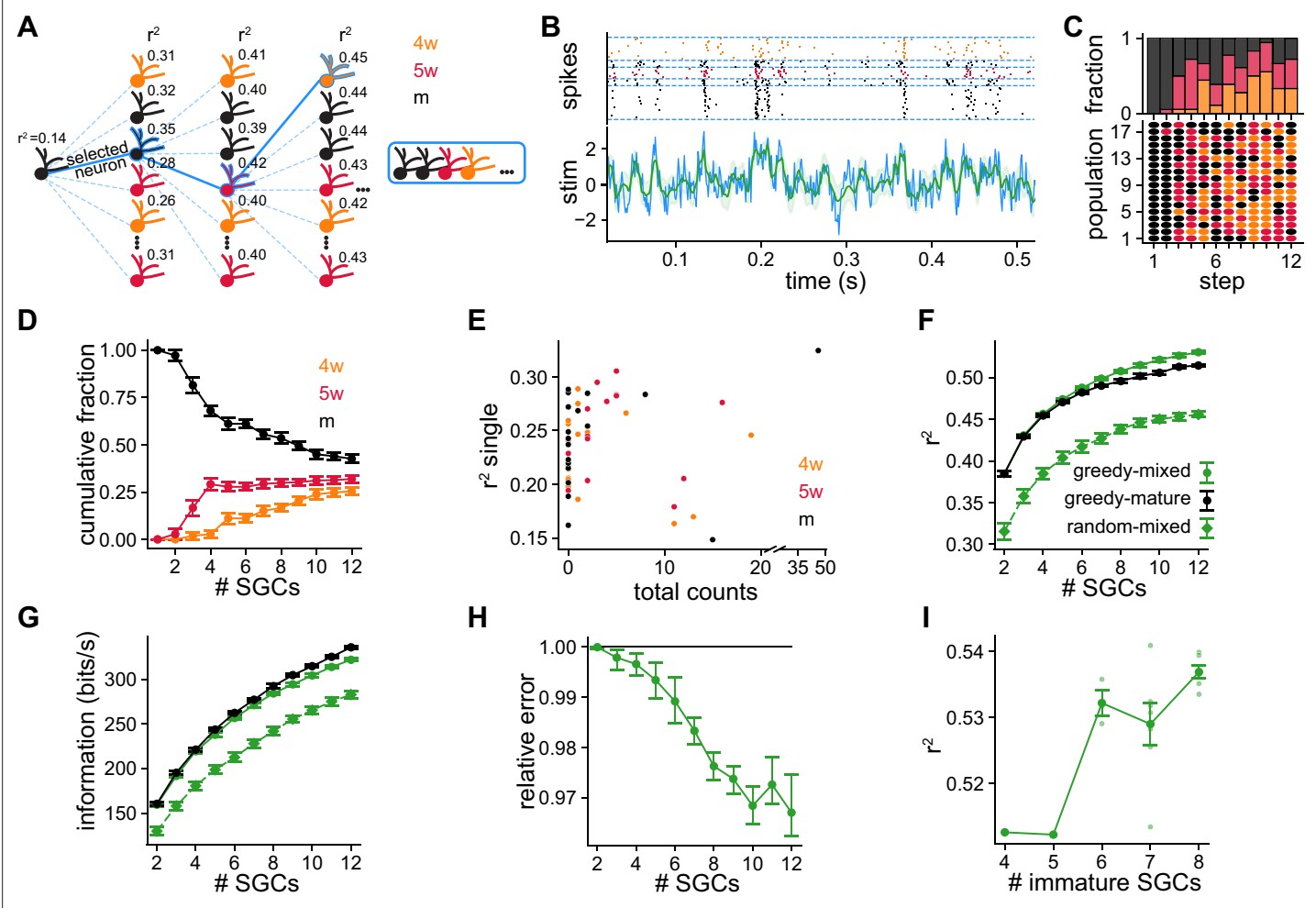

**Figure 4.** Greedy procedure used to build populations of simulated granule cells (SGCs) optimized for stimulus reconstruction. (**A**) Greedy procedure diagram: at each step, the SGC that optimizes stimulus reconstruction measured by average $r^2$ is chosen. (**B**) Decoding example using a population of five SGCs of different ages. Top: Raster plot showing the spike timings of each SGC separated by blue dashed lines. Each SGC contributes a different number of trials to equalize the total number of spikes. Bottom: The original stimulus (blue line) is shown together with the reconstructed stimulus (green line) and its uncertainty (shaded green) ($r^2 = 0.48$). (**C**) Bottom: Selected SGCs at each step. Rows are populations of SGCs selected with the greedy procedure. Dot color represents SGCs' ages. Top: Age fractions of selected SGCs at each step. (**D**) Average cumulative fraction for each age for increasing number of SGCs in the populations. Error bars indicate ±1 s.e.m. (**E**) Average $r^2$ value achieved by single SGCs vs. total number of times the SGC was selected. (**F, G**) Average over populations of (**F**) $r^2$ and (**G**) mutual information for increasing number of SGCs in the populations. Error bars indicate ±1 s.e.m. (**H**) Reconstruction mean squared error of greedy mixed-age populations relative to greedy mature populations computed by bootstrapping populations from both groups. Error bars indicate symmetric 95% c.i. (**I**) Average $r^2$ vs. number the of immature SGCs for populations of 12 SGCs. Spearman's correlation: $\rho = 0.67$, $p = 2.3 \times 10^{-3}$. Darker dots indicate averages and error bars indicate ±1 s.e.m.

The online version of this article includes the following figure supplement(s) for figure 4:

**Figure supplement 1.** Statistics of SGC selection.

**Figure supplement 2.** Greedy procedure with restricted mSGC selection in the first steps.

**Figure supplement 3.** Decoding stimuli with underlying theta oscillations.

pairs of SGCs (*Figure 3—figure supplement 3*), mSGCs were chosen almost exclusively among the full pool in the second step. Unexpectedly, from the third step onward, immature SGCs were chosen by the greedy procedure with increasing preference (*Figure 4C*). The fraction of immature SGCs in the populations increased at the expense of the mSGCs (*Figure 4D*). For populations of 12 SGCs, approximately 43% were mSGCs, 32% were 5wSGCs, and 25% were 4wSGCs. We found that 51% of all the SGCs were never chosen in any population by the greedy algorithm, 34.5% of the SGCs were selected between 1 and 10 times, and 14.5% were selected over 10 times (*Figure 4—figure*

*supplement 1*). This indicates that the final populations strongly differ from random selection, which would have each SGC selected 3.6 times on average. Furthermore, these results highlight the diversity of the final populations, since a large fraction of SGCs were selected a smaller number of times than the number of populations, hence these are not composed of the same neurons. Moreover, although some of the preferentially chosen SGCs achieved large $r^2$ values when used individually, some highly selected SGCs yielded relatively poor reconstructions by themselves (*Figure 4E*) as well as in pairs (*Figure 4—figure supplement 1*). These observations indicate that decoding performance is a result of a non-trivial synergy between the GCs composing a population, and optimal performance is not achieved by taking the best individual GCs.

The greedy procedure selected immature SGCs to improve stimulus reconstruction. Both $r^2$ and the mutual information steadily increased with the number of SGCs in the populations, and were clearly larger than those of populations of randomly selected SGCs (*Figure 4F, G*). This improvement in the decoding that resulted in larger values of $r^2$ can also be contrasted to single simulated neurons, with $r^2 \in (0.02, 0.12)$ (*Figure 3D* and *Figure 3B*), and with multiple trials of single simulated neurons $r^2 \in (0.15, 0.3)$, (*Figure 4E* and *Figure 3G*). Still, the greedy procedure performs a stepwise optimization, therefore choosing immature SGCs over mSGCs early in the construction could result in worse decoding performance for the final populations. We thus compared the decoding performance of the greedy mixed-age populations with exclusively mSGCs' populations, built following the greedy procedure and starting from the same pool of initial mSGCs. With increasing number of neurons, mixed-age populations achieved consistently larger $r^2$ values in comparison with exclusively mature ones (*Figure 4F*). This resulted in a steady average decrease in the relative reconstruction error, reaching a final relative improvement of 3% (*Figure 4H*). Moreover, we found a positive correlation between the $r^2$ of a population and the number of immature SGCs that it contains (*Figure 4I*). However, the populations of mSGCs achieved larger information values than the mixed-age ones (*Figure 4G*). The mutual information, unlike the $r^2$, is not quantified by comparing the decoded stimuli with the true stimuli. It is related to the uncertainty about the decoding and the correspondence between decoded and true stimuli, but not to whether it is a good approximation to it. In fact, a decoder could achieve perfect mutual information but result in a poor reconstruction by performing a perfectly scrambled one-to-one mapping of the true stimulus (*Schneidman et al., 2003*).

After 12 iterations, the greedy procedure produced populations with at least four immature cells (*Figure 4I*). We wondered whether a smaller fractions of immature cells could still produce a significant effect on decoding. To generate populations with smaller proportions of immature cells, we constrained the greedy procedure by selecting only mature cells in the first steps. Varying the number of steps for which we restrict the selection to mSGCs, we generate populations with a variable number of immature SGCs. For example, for populations of 12 SGCs, if we restrict selection to mSGCs during the first nine steps, we generate populations that can have between zero and three immature SGCs. Although a single immature SGC did not have a significant impact in a 12 SGCs' population, from 2 immature SGCs onward there was an improvement in decoding as reflected in growing $r^2$ (*Figure 4—figure supplement 2*). Conversely, the mutual information decayed with increasing number of immature SGCs.

Finally, we wondered if immature SGCs improve decoding performance when the stimulus contains a theta oscillation as it has been observed in hippocampal rhythms (*Pernía-Andrade and Jonas, 2014*). Thus, we recorded GCs' responses to a stimulus template containing theta oscillations (Methods) (*Figure 4—figure supplement 3*). We used the SRM fits obtained in *Figure 2* to simulate SGCs' responses using this same stimuli. We found that the model was able to extrapolate to this theta-modulated stimulus, generating responses that captured the data. In the presence of theta rhythms, stimulus decoding was also improved by the presence of immature SGCs (*Figure 4—figure supplement 3*).

Thus, while single mGCs perform better than immature ones, mixed-age populations improve stimulus reconstruction, suggesting that the diversity contributed by immature GCs could be beneficial for transmitting distinct properties of stimuli with rich temporal structure.

## Immature GCs in a population improve stimuli discrimination

We showed that the presence of immature GCs in a population can enhance stimulus reconstruction. The dentate gyrus is thought to play a key role in pattern separation (*Leutgeb et al., 2007*). Ablating

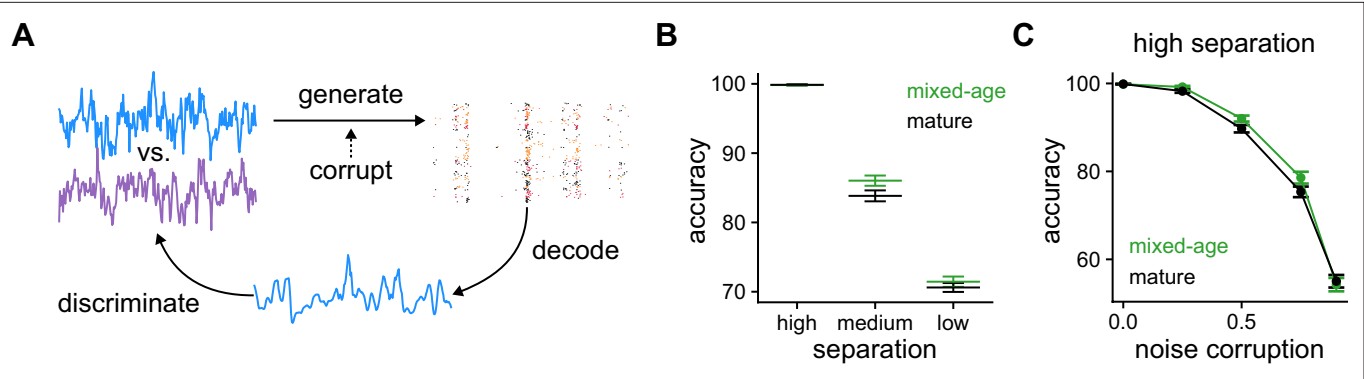

**Figure 5.** Pattern discrimination between pairs of fluctuating stimuli. (**A**) Diagram of the pattern discrimination procedure. (**B**) Discrimination accuracy achieved by mixed and exclusively mature populations for different degrees of separation between the two stimuli. Mann-Whitney U test high separation: $p = 0.36$, medium: $p = 1.0 \times 10^{-2}$, and low: $p = 0.19$. (**C**) Discrimination accuracy vs. level of noise corruption in the stimuli for high degree of separation. Mann-Whitney U test 0.25: noise $p = 3.6 \times 10^{-3}$, 0.5: $p = 2.3 \times 10^{-2}$, 0.75: $p = 6.1 \times 10^{-2}$, and 0.9: $p = 0.47$.

neurogenesis affects fine spatial discrimination, that is when the separation between patterns is small, but not when it is large (*Clelland et al., 2009*). Thus, we wondered whether a downstream region could benefit from reading the output of a mixed-age population to enhance discrimination of similar patterns. We designed a pattern separation task within the decoding framework by using it to discriminate between pairs of correlated stimuli (*Figure 5A*, Methods). We used mixed-age and mature-only populations of 10 SGCs that we constructed in the previous section. We generated pairs of correlated stimuli and then used one of them to generate spike trains. We then performed the decoding to obtain a stimulus reconstruction. To discriminate, we compared the errors between the reconstruction and each of the two stimuli, expecting that this discrimination task would become increasingly harder as the correlation between stimuli increases.

We then explored whether the populations could be used to correctly discriminate between the stimuli with different degrees of separation. We defined three degrees of separation (low, medium, and high) that correspond to different correlation values between the jointly generated stimuli (*Figure 5B*, Methods). For high separation the two stimuli are easily discriminated with almost 100% accuracy both by the mixed-age and mature-only populations. For low separation, stimuli are hard to discriminate and while the obtained accuracy is above chance, the differences between the mixed-age and mature-only populations are small. However, for medium degree of separation the mixed-age populations outperformed the mature-only at the discrimination task, suggesting that age heterogeneity in the dentate gyrus could also be leveraged to separate time-varying stimuli.

Finally, we tested whether immature SGCs could also improve discrimination if the stimulus is corrupted with noise. To do this, for each SGC we produced a different corrupted version of the stimulus used for generating spike trains. As now each spike train is generated with a corrupted version of the stimulus, correct discrimination will depend on the capability of the decoder to average out the noise in the population. For a fixed degree of separation between the pairs of stimuli, adding noise is detrimental to discrimination performance for the populations (*Figure 5C*). Notably, the mixed-age populations consistently outperform the mature-only for moderate amounts of noise. In summary, our results suggest that immature GCs in the dentate gyrus introduce a degree of heterogeneity at the population level that can be leveraged to reconstruct an incoming stimulus better and, at the same time, discriminate it from others.

## Discussion

Neurogenesis contributes a substantial number of GCs to the dentate gyrus (*Imayoshi et al., 2008*). Previous works have studied the electrophysiological properties of adult-born immature GCs in hippocampal slices (*Mongiat et al., 2009*; *Marín-Burgin et al., 2012a*; *Pardi et al., 2015*). Other works have addressed the impact of perturbing neurogenesis on behavior (*Clelland et al., 2009*; *Nakashiba et al., 2012*). However, in vivo studies that record from immature GCs are rare (*Danielson et al., 2016*), and the mechanisms by which neurogenesis influences coding are largely unknown. Here,

we approach this question combining measurements in hippocampal slices and statistical modeling to study the impact of age heterogeneity on stimulus encoding. We explore the idea that immature GCs could actively aid representation and discrimination of stimuli. We record the responses of dentate gyrus GCs of different ages to stimuli with rich temporal structure. The fluctuating stimuli we use produce reliable responses, allowing us to study their structure over time and across trials. We find that immature GCs produce more variable responses, less reproducible across trials, and less aligned with the stimulus. We then fit an SRM to capture the subthreshold membrane potential and spiking responses. We find that immature GCs integrate stimuli over longer time scales and exhibit weaker refractory effects. We show that these parameters can be used to discriminate GCs' ages. Decoding stimuli from the responses of single SGCs, we find that stimulus reconstruction and information improve with maturation. Unexpectedly, despite the worse individual performance of immature SGCs, we find that they aid decoding in a simulated population. Finally, we design a pattern separation task using our framework and show that immature SGCs enhance discrimination of highly correlated stimuli.

To mimic inputs with complex temporal structure, we use fluctuating current injections and build statistical models to investigate the differences between GCs of different ages. An alternative approach could be to stimulate with conductances using dynamic clamp (*Prinz et al., 2004*). However, conductance stimuli would not allow for the straightforward fitting of the GLM, since it assumes that the membrane potential is a linear convolution of the input. Here, we block inhibitory and excitatory transmission to control the current stimulus and investigate the interaction between the stimulus and the intrinsic properties of GCs of different ages. We observe that immature GCs produce generally less reliable responses, that are also less correlated with the stimulus and with the responses of other GCs. Immature GCs' responses are also more variable in vivo, exhibiting less spatial specificity when animals perform a spatial exploration task (*Danielson et al., 2016*). Immature GCs' responses are less controlled by inhibitory circuits than mGCs, spiking in response to weaker stimuli and with more variable timing (*Marín-Burgin et al., 2012a*; *Pardi et al., 2015*). In addition, although at 4 weeks immature GCs are contacting their postsynaptic targets, the connections are still maturing so there may still be differences in the strength of the synapses (*Toni et al., 2008*). Thus, the differences in reliability that we report here could be further increased by the network. Additionally, the amount of excitatory recurrence in the dentate gyrus is small (*Claiborne et al., 1986*) compared to other hippocampal areas like CA3, hence we do not incorporate recurrent connectivity when decoding with populations of GCs.

Using the SRM, we quantitatively characterize GCs' responses to the applied stimuli. Studying the SRM parameters, we observe that the membrane filters of immature GCs have larger amplitudes and longer time scales, their thresholds are lower, and they experience weaker refractory effects. Previous works using current steps have reported that immature GCs have larger input resistances and time constants, lower action potential thresholds, and smaller postspike potassium currents (*Mongiat et al., 2009*; *Yang et al., 2015*). These observations validate our approach and show that parameters obtained from the fit are linked to biophysical properties. Moreover, our SRM characterization reproduces the spiking responses, allowing us to further study the stimulus-response transformation performed by GCs. We find that immature GCs low reliability can be partly attributed to their longer time scales of stimulus integration.

The model-based decoding framework that we use allows us to study how GCs represent stimuli without making precise assumptions about the nature of the neural code. By decoding stimuli with single SGCs and populations, we find that reliable neurons generally achieve better decoding performance and that optimal populations combine diversity with homogeneity, using neurons with different properties while often including the same neuron multiple times. Similarly, a previous study reported similar observations using a model-based approach to explore the impact of intrinsic diversity in the olfactory bulb (*Tripathy et al., 2013*). Our study explores not only the impact of random cell-to-cell diversity in the GC population, but also the benefits of the structured heterogeneity introduced by neurogenesis.

Despite the fact that single mSGCs achieve better decoding performance than immature ones, immature SGCs are selected when we build populations that optimize stimulus reconstruction. In the hippocampus, the postsynaptic area is constantly receiving new synapses from 4-week-old immature cells. These new connections may continuously reshape input processing in the area. In the simulations,

mixed-age populations yield better stimulus reconstruction than populations consisting of exclusively mSGCs. Correspondingly, a previous study involving spatial exploration has shown that immature GCs are less spatially tuned but actively participate in context encoding and discrimination (*Danielson et al., 2016*). Additionally, they reported that immature GCs fire at higher rates in vivo. In this study, we equalize the firing rates of different neurons, yet a higher firing rate would further improve the decoding performance of immature GCs and increase their importance in population coding.

We explore the capability of populations of SGCs to perform pattern separation in the time domain, formulating the discrimination of time-varying signals on the scale of milliseconds. While mechanisms involving code expansion and lateral inhibition are important for pattern separation (*O'Reilly and McClelland, 1994*; *Cayco-Gajic and Silver, 2019*), adult-born neurogenesis has been implicated in discrimination of patterns that are very similar (*Clelland et al., 2009*). Recently, a study using perforant path stimulation studied pattern separation in different development-born hippocampal neurons but they only used pulsed stimulation (*Madar et al., 2019*). Despite immature GCs' variable responses, we find that mixed-age populations are able to discriminate between highly correlated stimuli better than exclusively mature populations, even in the presence of uncorrelated noise in the population. The noisier output of immature GCs that we observe could also help disrupt established memories, consistent with the hypothesis that neurogenesis induces forgetting of existing memories, facilitating the formation of new conflicting ones (*Epp et al., 2016*). Computational studies of the dentate gyrus that incorporate neurogenesis have usually focused on learning aspects (*Aimone et al., 2009*; *Wiskott et al., 2006*). Using networks that continually incorporate new GCs to encode novel information, these studies propose that neurogenesis could prevent interference between new and old memories and aiding their discrimination.

It is intriguing that, while immature cells are less reliable individually, they still contribute to decoding in a population and improve performance in a pattern separation task. We explored the properties of selected individual GCs and found that many immature GCs that were highly selected in the populations did not perform well individually in decoding the stimulus. Thus, this rules out simple explanations for these observations and points to non-trivial synergistic effects in the mixed population. It is possible that immature GCs' code aspects of the stimulus that are not represented in mature cells. This is an interesting open question for future work. In addition to individually contributing to decoding, immature neurons may modulate mGCs' activity improving stimulus reconstruction, for example by recruiting inhibitory circuits (*Temprana et al., 2015*; *Anacker and Hen, 2017*).

Our work adds evidence in favor of the idea that intrinsic heterogeneity is beneficial for population coding (*Padmanabhan and Urban, 2010*; *Tripathy et al., 2013*). Particularly, our study indicates that the intrinsic diversity of immature GCs contribute to the dentate gyrus could be, by itself, beneficial for stimulus representation and discrimination. A general mechanistic understanding of how the dentate gyrus leverages immature GCs to process incoming activity is missing. Our approach could be used and combined with other experimental procedures to guide future experiments and further advance this understanding.

## Methods
### Mice and slice preparation

Ascl1-CreERT2 mice (*Yang et al., 2015*) (Jackson Laboratory RRID IMSRJAX:012882) were crossed with CAG-floxStop-tdTomato (Jackson Laboratory RRID IMSRJAX:007914) mice to generate Ascl1-CreERT2-Tom mice. Mice were housed with a running wheel which is known to enhance adult hippocampal neurogenesis (*van Praag et al., 1999*). Adult mice of either sex were injected with tamoxifen at 6–8 weeks of age. Tamoxifen was delivered intraperitoneally in two 120 µg per mouse gram injections in 2 consecutive days to achieve indelible expression of Tom in newborn GCs. Mice were anesthetized and decapitated at 4 or 5 weeks after injection, depending on the desired age of the GCs. Experimental protocol (2020-03-NE) was evaluated by the Institutional Animal Care and Use Committee of the IBioBA-CONICET according to the Principles for Biomedical Research involving animals of the Council for International Organizations for Medical Sciences and provisions stated in the Guide for the Care and Use of Laboratory Animals. Brains were removed and placed into a chilled solution containing (mM) 110 choline chloride, 2.5 KCl, 2.0 $NaH_2PO_4$, 25.0 $NaHCO_3$, 0.5 $CaCl_2$, 7 $MgCl_2$, 20 dextrose, 1.3 sodium ascorbate, 0.6 sodium pyruvate. Acute slices 400 µm thick from

either hemisphere were cut transversally to the longitudinal axis in a vibratome. Hippocampal slices were then transferred to a chamber containing artificial cerebrospinal fluid (mM): 125 NaCl, 2.5 KCl, 2.3 NaH$_2$PO$_4$, 25 NaHCO$_3$, 2 CaCl$_2$, 1.3 MgCl$_2$, 1.3 Na+-ascorbate, 3.1 Na+-pyruvate, and 10 dextrose (315 mOsm). Slices were bubbled with 95% O$_2$/5% CO$_2$ and maintained at 30°C for >1 hr before experiments started.

## Electrophysiological recordings

Recorded immature neurons were visually identified by fluorescence. GCs used between 28 and 30 days post injection were labeled as 4w and GCs used between 34 and 36 days post injection were labeled as 5w. The mature population encompassed unlabeled neurons localized in the outer third of the GC layer (*Mongiat et al., 2009*; *Yang et al., 2015*). Whole-cell current-clamp recordings were made at room temperature, in the estimated range from 22°C to 26°C. We used microelectrodes (4–6 MΩ) filled with a potassium gluconate internal solution (in mM): 120 potassium gluconate, 4 MgCl$_2$, 10 HEPES buffer, 0.1 EGTA, 5NaCl, 20KCl, 4ATP-tris, 0.3 GTP-tris, and 10 phosphocreatine (pH = 7.3; 290 mOsm). Excitatory and inhibitory synaptic transmission were blocked pharmacologically with kynurenic acid (Sigma-Aldrich Cat K3375) and picrotoxin (Sigma-Aldrich Cat P1675). Recordings were obtained using Multiclamp 700B amplifiers (Molecular Devices), digitized using Digidata 1550 (Axon instruments), and acquired at 10 kHz onto a personal computer using the pClamp10 software (Molecular Devices). Input resistance was obtained from current traces evoked by a hyperpolarizing step of 10 mV. Series resistance was typically 10–20 MΩ, and experiments were discarded if higher than 40 MΩ. Before starting every protocol of stimulation, the resting membrane potential was kept at around –70 mV by passing a holding current. To compute the reported input resistance and time constants, we injected small current hyperpolarizing steps and performed exponential fits of the resulting membrane potential deviations. Current step amplitude was adapted to each neuron and resulted in a negative membrane potential deviation of 1–4 mV. All electrophysiological recordings are available as raw data from Dryad.

## Fluctuating stimulus

We based the design of the stimulus on in vivo recordings from the intact network (*Pernía-Andrade and Jonas, 2014*). These recordings show that GCs receive a wide range of frequencies, with a power spectrum that exhibits a power law decay. In vivo recordings have also reported a peak in the spectrum, indicating the presence of theta oscillations. Here, we did not include these oscillations in the stimuli because white noise, or noise with an exponentially decaying autocorrelation, produces better fits with GLMs (*Paninski, 2004*). We generated fluctuating stimuli from an Ornstein-Uhlenbeck process (*Gillespie, 1996*)

$$\frac{d\eta}{dt} = -\eta/\tau + \epsilon(t). \tag{1}$$

The process has zero mean, unit variance, and correlation time constant $\tau$. This process was implemented numerically with the discretization

$$\eta(t + \Delta t) = \eta(t)e^{-\Delta t/\tau} + \sqrt{1 - e^{-2\Delta t/\tau}}\mathcal{N}(0, 1) \qquad \eta(0) = \mathcal{N}(0, 1) \tag{2}$$

Throughout this work we used a time constant $\tau = 3$ ms that was much shorter than neuron membrane time constants. Both in experiments and simulations described below, neurons were injected with currents of the form $i(t) = \sigma\eta(t) + \mu$, where $\mu$ set the mean value of the current and $\sigma$ determined the amplitude of current fluctuations. Since GCs of different ages have markedly different input resistances, during experiments we adapted $\mu$ and $\sigma$ online for each individual GC, seeking to obtain a firing rate of at least 1 Hz.

## Theta stimulus

We generated stimuli with a theta rhythm using the autocovariance function

$$\rho(u) = \sigma^2 \cos(2\pi fu)e^{-u/\tau} \tag{3}$$

with $f = 8\,\text{Hz}$ and $\tau = 100\,\text{ms}$. Given discrete time bins $\Delta t$, we used the autocovariance function to generate a covariance matrix $C$, with $C_{ij} = \rho((i - j)\Delta t)$. By using the Cholesky decomposition of $C$ to find a matrix $L$, with $C = LL^T$, we can then generate stimuli with the desired autocovariance by performing the matrix-vector product $L\epsilon$, where $\epsilon$ is vector which components are independent and $\epsilon_i \sim \mathcal{N}(0, 1)$. An offset $\mu$ was added to set the mean value of the current stimuli.

## Coincidence ratio

A spike train $s(t)$ with spike times $\{t_1, t_2, ...t_s, ...\}$ can be represented as the sum of Dirac delta distributions

$$s(t) = \sum_{t_s} \delta(t - t_s). \tag{4}$$

For two spike trains $s$ and $s'$, we defined a coincidence between them as the co-occurrence of two spikes within a predefined time window. We introduced the number of coincident spikes $c(s, s')$, defined by the inner product $\langle s, s' \rangle$ between the spike trains (*Gerstner, 2014*; *Naud et al., 2011*),

$$c(s, s') = \langle s, s' \rangle = \int_0^T dt \int_{-\infty}^{\infty} du \int_{-\infty}^{\infty} du'\, K(u, u')s(t - u)s'(t - u'), \tag{5}$$

where $T$ is spike train duration and $K(u, u') = h_\Delta(u)\delta(u')$ is a kernel with $h_\Delta(u) = \Theta(u + \Delta/2)\Theta(u - \Delta/2)$ a rectangular function of width $\Delta$, $\Theta$ the Heaviside step function, and $\delta$ the Dirac delta distribution. To compute all the quantities introduced here, we always used the time window $\Delta = 8$ ms that is similar to the duration of an action potential. For this choice of kernel and the small time window used, the coincidence of a spike train $s$ with itself always counted the total number of spikes $n(s)$ in the train, $\langle s, s \rangle = n(s)$ for all spike trains considered. Given two different sets of $M$ spike trains $S = \{s^1, s^2, ..., s^M\}$ and $S' = \{s'^1, s'^2, ..., s'^M\}$, we define the coincidence ratio between them

$$C(S, S') = \frac{\frac{1}{M^2} \sum_{i,j=1}^{M} c(s^i, s'^j)}{(N(S) + N(S'))/2}, \tag{6}$$

where the numerator is the average number of coincidences between the two sets and $N(S) = \frac{1}{M} \sum_{i=1}^{M} n(s^i)$ is the average number of spikes in the trains of the set $S$. This coincidence ratio is the average number of coincidences between the sets normalized by the average number of spikes in their spike trains. For a single recording $S$ consisting of $M$ trials, we define the reliability as

$$R(S) = \frac{\frac{1}{M(M - 1)} \sum_{i \neq j}^{M} c(s^i, s^j)}{N(S)}, \tag{7}$$

the average number of coincidences between different spike trains in the recording normalized by the average number of spikes in its spike trains. Thus, this reliability quantifies the variability within a single set of spike trains.

## Spike response model

The SRM generates spikes from a Bernoulli probability distribution with an instantaneous rate that is determined from the subthreshold membrane potential. The subthreshold membrane potential is

$$v(t) = v_b + \int_0^t k(u)i(t - u)du + \sum_{t_s \in S} h_v(t - t_s) \tag{8}$$

where $v_b$ is a voltage bias and $k(t)$ is a membrane filter that is convolved with the stimulus $i(t)$. A voltage history filter $h_v(t)$ is added after every spike for the set of spike times $S$. The instantaneous rate or conditional intensity $\lambda(t)$ is defined from the subthreshold membrane potential as

$$\lambda(t) = \exp\left( \left( v(t) - v_{th} - \sum_{t_s \in S} h_{th}(t - t_s) \right)/\Delta v \right) \tag{9}$$

where $v_{th}$ is a fixed voltage threshold, $h_{th}(t)$ is the threshold deflection every time there is a spike, and $\Delta v$ is a voltage scale. Finally, a spike is produced at time $t$ with probability

$$P(\text{spike at } t) = 1 - \exp\left(-\lambda(t)\Delta t\right). \tag{10}$$

with a discretized time interval $\Delta t$.

## SRM fitting

To fit the SRM we expand the three filters $k(t)$, $h_v(t)$, and $h_{th}(t)$ as linear combinations

$$k(t) = \sum_{j=1}^{J} a_j f_j(t) \qquad h_v(t) = \sum_{l=1}^{L} b_l f_l(t) \qquad h_{th}(t) = \sum_{m=1}^{M} c_m f_m(t), \tag{11}$$

where the $f_j(t)$ are rectangular basis functions with support $[t_j, t_{j+1})$, that is $f_j(t) = 1$ if $t_j \le t < t_{j+1}$ and $f_j(t) = 0$ otherwise. We used the same bin size for all the rectangular functions of the same filter. For $k(t)$ we used 44 bins of 8 ms width from $t_1 = 0$ ms to $t_{44} = 344$ ms, for $h_v(t)$ we used 17 bins of 25 ms width from $t_1 = 25$ ms to $t_{17} = 425$ ms, and for $h_{th}(t)$ we used 18 bins of 25 ms width from $t_1 = 0$ ms to $t_{18} = 425$ ms. The predicted subthreshold voltage $\hat{v}(t)$ is then a linear combination of the parameters

$$\hat{v}(t) = v_b + \sum_{j=1}^{J}\left(\sum_{u=0}^{t} f_j(u)i(t-u)\right)a_j + \sum_{l=1}^{L}\left(\sum_{t_s \in S} f_l(t-t_s)\right)b_l = \boldsymbol{x}(t)^{\mathsf{T}}\boldsymbol{\theta}_{\text{sub}} \tag{12}$$

where $\boldsymbol{x}(t) \in \mathbb{R}^{1+J+L}$ is the vector

$$\boldsymbol{x}(t) = \left(1, \sum_{u=0}^{t} f_1(u)\, i(t-u), ..., \sum_{u=0}^{t} f_J(u)\, i(t-u), \right.$$
$$\left. \sum_{t_s \in S} f_1(t-t_s), ..., \sum_{t_s \in S} f_L(t-t_s)\right), \tag{13}$$

$\boldsymbol{x}(t)^{\mathsf{T}}$ denotes the transposed vector, and $\boldsymbol{\theta}_{\text{sub}} = (v_b, a_1, ..., a_J, b_1, ..., b_L)$ is a vector containing the subthreshold model parameters. Introducing the matrix $\boldsymbol{X} \in \mathbb{R}^{N \times 1 + J + L}$ of rows $\boldsymbol{x}(t)$, we can write

$$\hat{\boldsymbol{v}} = \boldsymbol{X}\boldsymbol{\theta}_{\text{sub}}. \tag{14}$$

Here, we distinguish the predicted $\hat{\boldsymbol{v}}$ and recorded $\boldsymbol{v}$ subthreshold membrane potentials, and introduce vectors to write in compact form the time dependence at discretized times, that is $\boldsymbol{v} = (v(0), v(\Delta t), ..., v(T))$ with $T = (N-1)\Delta t$. We determined $\boldsymbol{\theta}_{\text{sub}}$ from the recorded potential by using linear least squares to minimize the MSE,

$$\hat{\boldsymbol{\theta}}_{\text{sub}} = \underset{\boldsymbol{\theta}_{\text{sub}}}{\text{argmin}}\ \text{MSE}(\hat{\boldsymbol{v}}(\boldsymbol{\theta}_{\text{sub}}), \boldsymbol{v}) = (\boldsymbol{X}^{\mathsf{T}}\boldsymbol{X})^{-1}\boldsymbol{X}^{\mathsf{T}}\boldsymbol{v}, \tag{15}$$

after removing the first 25 ms of voltage after every spike.

A set of spike times $S$ defines a spike train $\boldsymbol{s} = (s(0), s(\Delta t), ..., s(T))$ with $s(t) = 1$ if there is a spike at time $t$ and $s(t) = 0$ otherwise. The joint probability to observe a spike train $\boldsymbol{s}$ as a function of the threshold parameters $\boldsymbol{\theta}_{\text{th}}$ given the stimulus $\boldsymbol{i}$ is the likelihood $P(\boldsymbol{s} \mid \boldsymbol{i}; \boldsymbol{\theta}_{\text{th}})$. The log-likelihood function is (**Gerstner, 2014**)

$$\log P(\boldsymbol{s} \mid \boldsymbol{i}; \boldsymbol{\theta}_{\text{th}}) = \sum_{t_s \in S} \log \lambda(t_s) - \int_0^T \lambda(t)dt. \tag{16}$$

Defining

$$\lambda(t) = \exp\left(\boldsymbol{y}(t)^{\mathsf{T}}\boldsymbol{\theta}_{\text{th}}\right)$$
$$\boldsymbol{y}(t) = \left(\hat{v}(t), -1, \sum_{t_s \in S} f_1(t-t_s), ..., \sum_{t_s \in S} f_M(t-t_s)\right) \tag{17}$$
$$\boldsymbol{\theta}_{\text{th}} = \Delta v^{-1}(1, v_{th}, c_1, c_2, ..., c_M)$$

with $\hat{v}(t)$ determined by **Equation 14**, the log-likelihood can be written as

$$\log P(s \mid i; \theta_{\text{th}}) = \sum_{t_s \in S} y(t_s)^\top \theta_{\text{th}} - \Delta t \sum_{t=0}^{T} \exp\left(y(t)^\top \theta_{\text{th}}\right). \tag{18}$$

This log-likelihood is concave on the parameters and we can find its global maximum using Newton's method to perform gradient ascent (**Gerstner, 2014**). We also added a term to the right-hand side of **Equation 18** of the form

$$-\alpha \sum_{m=1}^{M-1} (c_{m+1} - c_m)^2 \tag{19}$$

to impose a degree of smoothness determined by $\alpha$ over the coefficients of the filter $h_{th}(t)$. We used the value of $\alpha$ that yielded the largest log-likelihood on the validation data. In this way, each of the two steps of the fitting procedure captures a different aspect of the recording, optimizing a different function. The first step aims to capture the subthreshold membrane potential by minimizing the MSE between predicted and measured, while the second step aims to capture the spiking by maximizing the log-likelihood.

We evaluated our fits by computing the root MSE between the predicted and recorded subthreshold voltage, the log-likelihood and a coefficient $M_d$ which quantifies the degree of similarity between SRM generated and recorded sets of spike trains (**Naud et al., 2011**; **Pozzorini et al., 2015**). We reported the log-likelihood relative to the log-likelihood of a Poisson process of the same rate and normalized by the number of spikes,

$$L = \frac{1}{n(s)} \log_2 \left( \frac{P(s \mid i; \theta)}{P(n(s) \mid \lambda)} \right) \quad \text{with} \quad \log P(n(s) \mid \lambda) = n(s) \log \lambda - \lambda T. \tag{20}$$

The degree of similarity $M_d$ was computed using the coincidences with rectangular kernels,

$$M_d(S, S') = \frac{\frac{1}{M^2} \sum_{i,j=1}^{M} c(s^i, s'^j)}{\left(N(S)R(S) + N(S')R(S')\right)/2} \tag{21}$$

where $c(s, s')$ is given by **Equation 5**, $R(S)$ by **Equation 7**, and $N(S)$ is the average number of spikes in the trains of the set $S$. In contrast to the coincidence ratio, $M_d$ takes into account the reliability within each set of spike trains. The Python custom code developed to perform the model fitting and the decoding is available from Github.

## Decoding

Here, we describe how we decoded a single stimulus $\eta$ from the responses of multiple neurons. Given spike trains $s^j$ from GCs with SRM parameters $\theta^j$ produced by stimuli $i^j = \sigma^j \eta + \mu^j$, we found the maximum a posteriori estimate of $\eta$ as

$$\eta_{\text{MAP}} = \underset{\eta}{\text{argmax}} \log P(\eta \mid \{s^j\}; \{\theta^j\}) \tag{22}$$

with

$$\log P(\eta \mid \{s^j\}; \{\theta^j\}) = \sum_{j=1}^{M} \log P(s^j \mid \eta; \theta^j) + \log P(\eta) + \text{constant}(\eta). \tag{23}$$

We generated stimuli using **Equation 2**, resulting in

$$\log P(\eta) = -\frac{1}{2} \eta^\top \Sigma^{-1} \eta, \tag{24}$$

with

$$\Sigma_{ij}^{-1} = \frac{1}{1-\beta^2} \cdot \begin{cases} 1 & \text{if } i=j=1 \text{ or } i=j=N \\ 1+\beta^2 & \text{if } i=j \text{ with } 1 < i < N \\ -\beta & \text{if } |i-j|=1 \\ 0 & \text{otherwise,} \end{cases} \tag{25}$$

and $\beta = e^{-\Delta t/\tau}$. The log-posterior of *Equation 23* with the Gaussian prior of *Equation 24* is concave on $\eta$, so we found its global maximum using Newton's method to perform gradient ascent (*Pillow et al., 2011*). Once we found $\eta_{\mathrm{MAP}}$, approximating the posterior by a Gaussian as in *Pillow et al., 2011*, $P(\eta \mid \{s^j\}; \{\theta^j\}) \approx \mathcal{N}(\eta_{\mathrm{MAP}}, C)$ we used

$$C^{-1} = -\nabla_\eta \big( \nabla_\eta \log P(\eta \mid \{s^j\}; \{\theta^j\}) \big) \tag{26}$$

to determine the covariance matrix of the approximation and quantify uncertainty (*Pillow et al., 2011*). We used the squared root of the diagonal of $C$ to quantify the uncertainty in the estimation as a function of time and the determinant of $C$ to compute the mutual information. The matrix $C^{-1}$ is positive semi-definite, symmetric, and banded. We used these properties and implemented the Cholesky decomposition from SciPy (*Virtanen et al., 2020*) to compute the optimization updates, the diagonal of $C$ and the determinant of $C$ efficiently, and saving memory.

Given a set of GCs indexed by $j$, we sampled multiple stimuli from *Equation 2*. For each stimulus we used the currents $i^j = \sigma^j \eta + \mu^j$ to generate spike trains and performed the decoding. We quantified the reconstruction error using the coefficient of determination,

$$r^2 = 1 - \frac{\mathrm{MSE}(\eta, \eta_{\mathrm{MAP}})}{\mathrm{Var}(\eta)} \tag{27}$$

averaged over all sampled $\eta$. We obtained the mutual information between the spike trains $\{s^j\}$ and the stimulus $\eta$,

$$I(\eta; \{s^j\}) = H(\eta) - H(\eta \mid \{s^j\}) \tag{28}$$

using all the decoded stimuli by computing

$$H(\eta \mid s) = \int_{s'} P(s') H(\eta \mid s = s') \approx \frac{1}{K} \sum_{k=1}^{K} H(\eta \mid \{s\} = \{s\}^k). \tag{29}$$

We used the Gaussian prior and Gaussian approximation of the posterior to compute the Gaussian entropies

$$H(\eta) = \log \sqrt{(2\pi e)^N |\Sigma|} \quad \text{and} \quad H(\eta \mid s = s') \approx \log \sqrt{(2\pi e)^N |C|} \tag{30}$$

where $|\cdot|$ is the matrix determinant.

Where we sought to compensate for differences in firing rates, we used a different number of trials for each GC to equalize the number of spikes that each one contributes to the decoding. To obtain $n$ spikes on average from a GC with firing rate $\lambda$ in a simulation of duration $T$, we used $\mathrm{round}(n/(\lambda T))$ trials.

To compute the metrics in *Figure 3* we repeated the decoding at least 100 times. To compute the metrics in *Figure 3—figure supplement 3* we repeated the decoding at least 50 times. To build the populations in *Figure 4* we repeated the decoding at least 20 times for each possible population that we tried.

## Pattern discrimination

We generated pairs of correlated stimuli $\eta(t) = (\eta_1(t), \eta_2(t))$ following

eLife Research article

Computational and Systems Biology | Neuroscience

$$\boldsymbol{\eta}(t + \Delta t) = \boldsymbol{\eta}(t)e^{-\Delta t/\tau} + \sqrt{1 - e^{-2\Delta t/\tau}}\mathcal{N}(\mathbf{0}, \boldsymbol{K}) \qquad \boldsymbol{\eta}(0) = \mathcal{N}(\mathbf{0}, \boldsymbol{K})$$

$$\text{with} \quad \mathbf{0} = (0, 0) \quad \text{and} \quad \boldsymbol{K} = \begin{pmatrix} 1 & \rho \\ \rho & 1 \end{pmatrix} \tag{31}$$

with $0 \le \rho < 1$. The two processes have mean value 0, variance 1, time constant $\tau = 3$ ms, and correlation $\langle \eta_1(t)\eta_2(t) \rangle = \rho$. Given the time scales of different filters of the neurons (~ 100 ms), we decided to use 10 s stimuli, which was sufficiently long to be in a regime where the decoding performance and the pattern discrimination accuracy would not change with stimulus duration. We used only $\eta_1(t)$ to generate spike trains and perform the decoding. We computed the MSE between the reconstruction and both stimuli $\eta_1(t)$ and $\eta_2(t)$. If $\text{MSE}(\boldsymbol{\eta}_{\text{MAP}}, \boldsymbol{\eta}_1) < \text{MSE}(\boldsymbol{\eta}_{\text{MAP}}, \boldsymbol{\eta}_2)$ then the patterns were correctly discriminated and incorrectly otherwise. By sampling multiple pairs $\boldsymbol{\eta}_1$, $\boldsymbol{\eta}_2$ we quantified the accuracy of the discrimination task as the percentage of correctly discriminated stimuli. Since we expected the impact of adult-born neurons to be important only for relatively large correlation coefficient values, for quantitation, we introduced three degrees of separation defined as low ($\rho$=0.9997), medium ($\rho$=0.999), and high ($\rho$=0.99). For each population of neurons and each degree of separation and noise corruption in *Figure 5*, we performed the decoding and stimulus discrimination 200 times.

## Acknowledgements

We thank Emilio Kropff for valuable comments on the manuscript, and members of the Marin Burgin and Morelli labs for fruitful discussions. This work was supported by IDRC 108878 and ANPCyT grants PICT 2015 0634 and PICT 2018 0880 awarded to AMB, ANPCyT grants PICT 2017 3753 and PICT 2019 0445 awarded to LGM, and FOCEM-Mercosur (COF 03/11) awarded to IBioBA.

## Additional information

### Funding

| Funder | Grant reference number | Author |
|---|---|---|
| Agencia Nacional de Promoción Científica y Tecnológica | PICT 2015 0634 | Antonia Marin-Burgin |
| Agencia Nacional de Promoción Científica y Tecnológica | PICT 2018 0880 | Antonia Marin-Burgin |
| Agencia Nacional de Promoción Científica y Tecnológica | PICT 2017 3753 | Luis G Morelli |
| Agencia Nacional de Promoción Científica y Tecnológica | PICT 2019 0445 | Luis G Morelli |
| International Development Research Centre | IDRC108878 | Antonia Marin-Burgin |
| FOCEM-Mercosur | COF 03/11 | Diego M Arribas Antonia Marin-Burgin Luis G Morelli |

The funders had no role in study design, data collection and interpretation, or the decision to submit the work for publication. Open access funding provided by Max Planck Society.

### Author contributions

Diego M Arribas, Conceptualization, Software, Formal analysis, Investigation, Visualization, Methodology, Writing - original draft, Writing - review and editing; Antonia Marin-Burgin, Luis G Morelli, Conceptualization, Resources, Formal analysis, Supervision, Funding acquisition, Methodology, Writing - original draft, Project administration, Writing - review and editing

Author ORCIDs
Diego M Arribas http://orcid.org/0000-0002-6190-4770
Antonia Marin-Burgin http://orcid.org/0000-0003-0684-9796
Luis G Morelli http://orcid.org/0000-0001-5614-073X

## Ethics

Experimental protocol (2020-03-NE) was evaluated by the Institutional Animal Care and Use Committee of the IBioBA-CONICET according to the Principles for Biomedical Research involving animals of the Council for International Organizations for Medical Sciences and provisions stated in the Guide for the Care and Use of Laboratory Animals.

## Decision letter and Author response

Decision letter https://doi.org/10.7554/eLife.80250.sa1
Author response https://doi.org/10.7554/eLife.80250.sa2

## Additional files

### Supplementary files
• MDAR checklist

### Data availability

The data generated in this study is publicly available at Dryad, https://doi.org/10.5061/dryad.73n5tb309. Custom code produced and used in the study is available at Github, https://github.com/diegoarri91/iclamp-glm copy archived at *Arribas, 2023*.

The following dataset was generated:

| Author(s) | Year | Dataset title | Dataset URL | Database and Identifier |
|---|---|---|---|---|
| Arribas DM, Marin-Burgin A, Morelli LG | 2022 | Adult-born granule cells improve stimulus encoding and discrimination in the dentate gyrus | https://dx.doi.org/10.5061/dryad.73n5tb309 | Dryad Digital Repository, 10.5061/dryad.73n5tb309 |

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
