## [Editor Report]

The paper by Arribas et al. examines the coding properties of adult-born granule cells in the hippocampus at both the single cell and network level. This paper is of interest to the hippocampal and computational neuroscience fields because it provides a framework for understanding how adult-born granule cells in the hippocampus contribute to network processing. The paper contains interesting ideas, such as the analysis of input-output transformation by spike response models and the establishment of "greedy networks", and the conclusions drawn are supported by the data.

---

## [Decision Letter]

**Decision letter after peer review:**

[Editors’ note: the authors submitted for reconsideration following the decision after peer review. What follows is the decision letter after the first round of review.]

Thank you for submitting the paper "Adult-born granule cells improve stimulus encoding and discrimination in the dentate gyrus" for consideration by *eLife*. Your article has been reviewed by 3 peer reviewers, and the evaluation has been overseen by a Reviewing Editor and a Senior Editor. The following individual involved in the review of your submission has agreed to reveal their identity: Christoph Anacker (Reviewer #3).

Comments to the Authors:

We are sorry to say that, after consultation with the reviewers, we have decided that this work will not be considered further for publication by *eLife*.

This paper is of potential interest to both the hippocampal and computational neuroscience fields because it provides a framework for understanding how adult-born granule cells in the hippocampus contribute to network processing. It contains novel interesting ideas, such as the analysis of input-output transformation by SRM models and the establishment of "greedy networks". However, the degree of novelty is limited. That the adult-born granule cells have a slower integration time constant is known. Further, not all major conclusions are sufficiently supported by the data. The paper demonstrates that mixed networks show better encoding performance than pure networks, but the differences are small and only visible with specific performance metrics. Intuitive explanations are not provided. The reviewers found the approach intriguing and with new analysis, simulation, and experiment, it can form the basis of a future new submission.

*Reviewer #1 (Recommendations for the authors):*

As a general comment, there were a few points where I was not sure if an analysis was done using recorded spike trains or simulated spike trains. It would be helpful to clarify these points. For instance, the diagram in Figure 3A makes me think that this analysis is of simulated spike trains, but the caption states "Neurons encode a stimulus in spike trains that can be used to estimate the stimulus that produced them." Some of the comments regarding Figure 4 may reflect my confusion here.

In the Figure 1 discussion on page 5, you say, "this [higher coincidence ratios in older vs younger pairs of GCs] could be a consequence of individual immature GCs producing less reproducible responses." To what extent is this from lower reliability vs. differences in response functions (i.e., differences in the SRM parameters between the groups)? I may have missed it, but it seems you could say more on this point from the analysis in Figure 2.

Figure 4B: missing labels for green and blue curves. I'm a little unsure of how it is that the reconstruction is so good when the spike raster has so few spikes, such as between 250ms and 350 ms. It does appear that 4w cells may fire more in that period than m-cells. Is this part of why including 4w cells is helpful? In other words, the "nontrivial synergy" (p. 11) arises from selecting a group of cells in which at least some of the cells are firing at a given time?

In Figure 4, the results of decoding from pseudo-populations of GCs are shown. Something missing for interpretation here is what the ceiling of ideal performance is. In other words, is an r^2 of slightly more than 0.5 "good"? Supposing you took the parameters from models of GCs, then simulated a pseudopopulation using the SRM, and then used these spikes for the decoding procedure. How well would the model-based decoding work with these simulated spike trains?

The authors find that including the mixed ages in the decoding pool can improve the reconstruction, though somehow the information is not quite as high. I do not understand why this is. Is it because of spike count normalization? Or a subtlety in the information calculation?

Regarding the information calculations:

1. Please include an explanation of how the Gaussian approximation for information calculations was justified, and when it may not be (for instance, due to very low spike counts).

2. In Figure 4G, it'd be useful to know the bits per second or bits per spike, rather than bits.

Also in Figure 4, I'm curious about the composition of the multi-cell groups. What are the typical time constants (k, Figure 2I) or other model parameters of these greedy-optimized cell groups? Does the improvement in performance arise from having a broad sample of time constants, irrespective of whether it's a 4w/5w/m cell? Put another way, is the gap between m-groups and mixed-groups accounted for by drawing one of the ~2 cells in the 4w or in the 5w groups that happen to have much longer k time constants than any of the m-cells?

On Figure 5: (sequence discrimination) I cannot find how long the stimuli used for pattern discrimination are. How does pattern discrimination depend on stimulus length? Also, I missed a point made in the methods regarding the sequence design and rho. I would expect that more correlated stimuli would be less separated, but in "Pattern Discrimination," it says that rho is the correlation between eta_1 and eta_2, and low rho is a low degree of separation (rho = 0.99, low; rho = 0.9997, high).

Another question, which is more for discussion than further analysis, is how this heterogeneity relates to integrating new cells into a network. A week is a relatively short timeframe. Is the picture that some optimal downstream decoder is constantly seeking out these 4-week-old cells to improve decoding?

*Reviewer #3 (Recommendations for the authors):*

I have only one additional comment on this otherwise excellent study:

The authors find that while immature granule cells are not as reliable in stimulus encoding as mature granule cells, populations containing immature neurons perform better in stimulus reconstruction. The authors discuss the potential benefit of cellular diversity in stimulus reconstruction due to immature neurons potentially encoding different stimulus properties depending on their age. It should also be discussed that in addition to contributing to encoding themselves, the immature neurons may have a modulatory role on the mature granule cells that may improve stimulus reconstruction of the granule cell population when immature neurons are included.

---

## [Author Response]

[Editors’ note: The authors appealed the original decision. What follows is the authors’ response to the first round of review.]

Reviewer #1 (Recommendations for the authors):As a general comment, there were a few points where I was not sure if an analysis was done using recorded spike trains or simulated spike trains. It would be helpful to clarify these points. For instance, the diagram in Figure 3A makes me think that this analysis is of simulated spike trains, but the caption states "Neurons encode a stimulus in spike trains that can be used to estimate the stimulus that produced them." Some of the comments regarding Figure 4 may reflect my confusion here.

We realize that in the original version we used the term neuron too broadly, and this was confusing. In particular, Figure 3A is a diagram generated with simulated spike trains, meant for illustrative purposes. However, the decoding procedure can be performed using either experimental or simulated spike trains, and we take advantage of this to validate the approach. Figure 3B,C shows experimental recordings and results, while the rest of Figure 3 uses simulations, and in Figure 3—figure supplement 1 we compare both, showing that decoding performance for single neurons was similar when using the recordings or the simulations.

We understand how this point could have been unclear. To distinguish between recordings and simulated spike trains without ambiguity, in the revised manuscript, we introduced a clear terminology: Granule Cell (GC) vs. Simulated Granule Cell (SGC).

In the Figure 1 discussion on page 5, you say, "this [higher coincidence ratios in older vs younger pairs of GCs] could be a consequence of individual immature GCs producing less reproducible responses." To what extent is this from lower reliability vs. differences in response functions (i.e., differences in the SRM parameters between the groups)? I may have missed it, but it seems you could say more on this point from the analysis in Figure 2.

We agree with the reviewer that this sentence may be misleading in hinting a causality relation between the change in reliability and the observed difference in coincidence ratios. Actually, as the reviewer points out, we do use this as a motivation to develop the model in the following section.

Thus, in the revised manuscript we reworded the misleading sentence to: “This could be related to individual immature GCs producing less reproducible responses, which can be quantified by the reliability.”

Figure 4B: missing labels for green and blue curves. I'm a little unsure of how it is that the reconstruction is so good when the spike raster has so few spikes, such as between 250ms and 350 ms. It does appear that 4w cells may fire more in that period than m-cells. Is this part of why including 4w cells is helpful? In other words, the "nontrivial synergy" (p. 11) arises from selecting a group of cells in which at least some of the cells are firing at a given time?

Concerning the Figure 4B missing labels, in the caption we write for this panel: “Bottom: The original stimulus (blue line) is shown together with the reconstructed stimulus (green line) and its uncertainty (shaded green).”

Concerning the reconstruction, note that in Figure 3B the presence of single spikes is already enough to generate bumps in the decoded stimulus that match upward stimulus deviations. Indeed, in the presence of only a few spikes the improvement is remarkable.

Concerning 4w cells firing at different times than mature cells, we agree with the reviewer that 4w cells may contribute to coding through their more dispersed firing, given that we can see clear differences in the accuracy of the decoding, in the absence or presence of spikes. For example, in Figure 4B the first peak before 300 ms is captured with a few spikes, while the first peak after 300 ms is not, due to the absence of spikes. By analyzing single neurons and pairs, we showed that immature neurons are less reliable (Figure 1J,I), which already indicates a larger “spreading” of spike timings.

To further explore the reviewer's suggestions, we analyzed the coincidence ratio of mature vs. immature neurons, new data in panel C of Figure 1—figure supplement 2. We found that pairs of mixed-age GCs that contain immature 4wGCs neurons produce smaller coincidence ratios than pairs of mature neurons alone, further indicating that 4wGCs spike at different timings than mature neurons. We comment on this result in the last paragraph of the section describing Figure 1.

In Figure 4, the results of decoding from pseudo-populations of GCs are shown. Something missing for interpretation here is what the ceiling of ideal performance is. In other words, is an r^2 of slightly more than 0.5 "good"? Supposing you took the parameters from models of GCs, then simulated a pseudopopulation using the SRM, and then used these spikes for the decoding procedure. How well would the model-based decoding work with these simulated spike trains?

What the reviewer suggests is exactly what we are doing in Figure 4, the spike trains used for decoding are indeed simulated. We think that one aspect of this comment is also related to the confusion between real and simulated neurons, see our reply to the first comment by the reviewer. As we stated above, in the revised manuscript we clarified this point across the manuscript to avoid these confusions by introducing the term Simulated Granule Cell (SGC).

We use r^2 to quantify the error because it has the “intuitive” interpretation of being the fraction of the variance explained by the decoded signal. In other words, this quantity captures the proportion of the variation in the stimulus that is predictable with the decoding. In the revised manuscript, we added this more intuitive explanation when we introduce this quantity to clarify its meaning.

Concerning the meaning of r^2 values, note that:

(i) r^2 of single simulated neurons are in the range of (0.02, 0.12) Figure 3D, (ii) multiple trials of single simulated neurons are in the range (0.15, 0.3) Figure 4E, (iii) while simulated populations of up to 12 neurons show r^2 above 0.5 Figure 4F.

For reference, we can see how reconstructions look like with

(i)single simulated neurons (Figure 3B),(ii)with multiple trials of single simulated neurons (Figure 3G) and (iii) five simulated neurons (Figure 4B).

This comparison shows that r^2 values do reflect an improvement in the decoding as appreciated by visual inspection of these reconstructions. However, we think that this information was scattered across figures and it was difficult to make the connections. Thus, in the revised manuscript we included a few sentences in the text to bring this comparison up front more clearly. Furthermore, we now report the r^2 values for all the decoding examples that we show in the manuscript, panels Figure 3B and 3G and Figure 4B.

We hope that with these changes we have made it more clear what r^2 values mean, and that an improvement from single neuron values in the range (0.15 , 0.30) to values above 0.5 the improvement is actually remarkable.

The authors find that including the mixed ages in the decoding pool can improve the reconstruction, though somehow the information is not quite as high. I do not understand why this is. Is it because of spike count normalization? Or a subtlety in the information calculation?

This is an interesting point. Note that the mutual information, unlike the r^2, is not quantified by comparing the decoded stimuli with the true stimuli. It’s related to the uncertainty about the decoding and the correspondence between decoded and true stimuli, but not to whether it is a good approximation to it. As pointed out by [Schneidman et al. 2003], a decoder could achieve perfect mutual information but result in a poor reconstruction by performing a perfectly scrambled one-to-one mapping of the true stimulus. The result of single mature neurons being overall better for decoding seems to generalize to the pseudo-populations we build when we look at the information but not when we look at the accuracy of the estimation as measured by r^2.

We agree that this is not straightforward. In the revised manuscript we added a few sentences to clarify this point.

Regarding the information calculations:1. Please include an explanation of how the Gaussian approximation for information calculations was justified, and when it may not be (for instance, due to very low spike counts).

The Gaussian approximation in the context of model based decoding was proposed by (Pillow et al. 2011). In the revised manuscript we point to this reference when we introduce it.

(Pillow et al. 2011) Jonathan W. Pillow, Yashar Ahmadian, and Liam Paninski. “Modelbased decoding, information estimation, and change-point detection techniques for multineuron spike trains”. In: Neural Computation 23.1 (2011), pp. 1–45.

2. In Figure 4G, it'd be useful to know the bits per second or bits per spike, rather than bits.

Thank you for the suggestion. In the revised version we plotted bits per second as suggested. We did this in Figure 4G, Figure 3E, F, H, J, Figure 3 Supp. 3 C, and Figure 3 Supp. 2 D. In Figure 3 Supp. 1 B we decided to keep the bits units, since the point is to make a comparison and we thought that this was more straightforward as is.

Also in Figure 4, I'm curious about the composition of the multi-cell groups. What are the typical time constants (k, Figure 2I) or other model parameters of these greedy-optimized cell groups? Does the improvement in performance arise from having a broad sample of time constants, irrespective of whether it's a 4w/5w/m cell? Put another way, is the gap between m-groups and mixed-groups accounted for by drawing one of the ~2 cells in the 4w or in the 5w groups that happen to have much longer k time constants than any of the m-cells?

This is a very good suggestion. In the revised manuscript we incorporated an additional analysis where we study the decoding performance of pseudo-populations that are built constraining the number of immature neurons in them, see new Figure 4—figure supplement 2. We discuss the approach and results after discussing Figure 4 in the main text.

On Figure 5: (sequence discrimination) I cannot find how long the stimuli used for pattern discrimination are. How does pattern discrimination depend on stimulus length? Also, I missed a point made in the methods regarding the sequence design and rho. I would expect that more correlated stimuli would be less separated, but in "Pattern Discrimination," it says that rho is the correlation between eta_1 and eta_2, and low rho is a low degree of separation (rho = 0.99, low; rho = 0.9997, high).

The duration of the stimulus is 10 seconds. In the revised manuscript we stated this in the Methods section, where we describe the implementation of the pattern separation task of Figure 5.

Note that given the timescales of the different filters of the neurons (~100 ms), we considered that 10 s was sufficiently long to be in a regime where the decoding performance and the pattern discrimination accuracy would not change with stimulus duration. In the revised manuscript we stated this argument explicitly together with the chosen stimuli duration.

As pointed out by the reviewer, the separation labels were inverted in the original version of the manuscript. In the revised version we corrected this in the Methods section as: low separation (ρ = 0.9997), medium separation (ρ = 0.999) and high separation (ρ = 0.99).

Another question, which is more for discussion than further analysis, is how this heterogeneity relates to integrating new cells into a network. A week is a relatively short timeframe. Is the picture that some optimal downstream decoder is constantly seeking out these 4-week-old cells to improve decoding?

In the context of adult neurogenesis, this 4-week window appears to be critical. At 4 weeks, these adult-born neurons are already connected to the postsynaptic neurons and are more active than older neurons. These features last about two weeks, where they become indistinguishable from mature neurons. Furthermore, removing the neurons of this age results in specific behavioral impairments. In the introduction we describe the data from the field that identifies this critical 4 week window for adult born neurons.

We think that rather than seeking out 4 week old cells, the postsynaptic area is constantly receiving new synapses from 4 weeks old immature cells that shape input processing in the area. In the revised manuscript we elaborate this idea in the discussion when we address the decoding.

Reviewer #3 (Recommendations for the authors):I have only one additional comment on this otherwise excellent study:The authors find that while immature granule cells are not as reliable in stimulus encoding as mature granule cells, populations containing immature neurons perform better in stimulus reconstruction. The authors discuss the potential benefit of cellular diversity in stimulus reconstruction due to immature neurons potentially encoding different stimulus properties depending on their age. It should also be discussed that in addition to contributing to encoding themselves, the immature neurons may have a modulatory role on the mature granule cells that may improve stimulus reconstruction of the granule cell population when immature neurons are included.

We thank the reviewer for the positive and encouraging comments about our work. We added a comment in the discussion about the modulatory role of immature neurons together with corresponding references.